# Induced cortical tension restores functional junctions in adhesion-defective carcinoma cells

Shoko Ito[1], Satoru Okuda[2], Masako Abe[3], Mari Fujimoto[3], Tetsuo Onuki[3], Tamako Nishimura[1,4] & Masatoshi Takeichi [1]

Normal epithelial cells are stably connected to each other via the apical junctional complex (AJC). AJCs, however, tend to be disrupted during tumor progression, and this process is implicated in cancer dissemination. Here, using colon carcinoma cells that fail to form AJCs, we investigated molecular defects behind this failure through a search for chemical compounds that could restore AJCs, and found that microtubule-polymerization inhibitors (MTIs) were effective. MTIs activated GEF-H1/RhoA signaling, causing actomyosin contraction at the apical cortex. This contraction transmitted force to the cadherin-catenin complex, resulting in a mechanosensitive recruitment of vinculin to cell junctions. This process, in turn, recruited PDZ-RhoGEF to the junctions, leading to the RhoA/ROCK/LIM kinase/cofilin-dependent stabilization of the junctions. RhoGAP depletion mimicked these MTI-mediated processes. Cells that normally organize AJCs did not show such MTI/RhoA sensitivity. Thus, advanced carcinoma cells require elevated RhoA activity for establishing robust junctions, which triggers tension-sensitive reorganization of actin/adhesion regulators.

[1] Laboratory for Cell Adhesion and Tissue Patterning, RIKEN Center for Developmental Biology, 2-2-3 Minatojima-Minamimachi, Chuo-ku, Kobe 650-0047, Japan. [2] Laboratoty for In Vitro Histogenesis, RIKEN Center for Developmental Biology, 2-2-3 Minatojima-Minamimachi, Chuo-ku, Kobe 650-0047, Japan. [3] Seed Compounds Exploratory Unit for Drug Discovery Platform, Drug Discovery Platforms Cooperation Division, RIKEN Center for Sustainable Resource Science, 2-1 Hirosawa, Wako 351-0198, Japan. [4] Present address: Nara Institute of Science and Technology, Ikoma 630-0192, Japan. Correspondence and requests for materials should be addressed to Nara.Institute.of.Science.and.Technology, Ikoma 630-0192JapanM.T. (email: takeichi@cdb.riken.jp)

One of the most important challenges in cancer treatment is to control metastasis[1]. Although many factors are thought to promote metastasis, histological abnormalities, such as loss of cell polarity and defective cell–cell adhesion are frequently observed in invasive tumors[2–4], and such abnormalities are thought to enhance cancer cell dissemination[5]. Our knowledge of how intercellular adhesion is impaired in tumor cells is still limited; however, normal epithelial cells develop the apical junctional complex (AJC)[6, 7], which consists of tight junction (TJ) and zonula adherens (ZA). A major molecular constituent of ZA is the E-cadherin adhesion receptor, whose cytoplasmic domain binds p120-catenin and β-catenin; β-catenin further binds αE-catenin, leading to formation of the cadherin-catenin complex (CCC)[8]. Although the CCC is generally important for cell–cell adhesion, the AJC plays a specific role in epithelial formation[9, 10]. The AJC associates with circumferential actomyosin cables via αE-catenin and other factors[11], and contraction of these cables produces tension over the AJC. This force is important for defining epithelial architecture[8, 12]. Actomyosin contraction is evoked by the RhoA-ROCK pathway. RhoA is activated by guanine nucleotide-exchange factors (GEFs) and inhibited by GTPase-activating proteins (GAPs)[13]. Some GEFs and GAPs are involved in junction regulation[14].

In human cancers, downregulation of E-cadherin correlates with invasive states[15–18]. Curiously, however, some colon carcinoma lines, such as Colo205 and HT29, express the core components of the CCC yet fail to organize normal junctions. Intriguingly, these cells are able to reorganize normal-looking junctions when treated with various factors[19–22], suggesting that their ability to organize the junctions is physiologically impaired. In the present study, we explored what are defective in such carcinoma cells through a bias-free screening of chemical compounds for their ability to restore normal junctions. We found that microtubule-polymerization inhibitors are dramatically effective. These inhibitors upregulated RhoA, consequently inducing actomyosin-mediated cortical contraction, which in turn led to a tension-dependent junctional reorganization. Carcinoma cells that normally form junctions did not respond to microtubule inhibitors in these ways. Thus, we report an unusual sensitivity of adhesion-defective carcinoma cells to microtubule inhibitors, and molecular mechanisms underlying the rebuilding of robust junctions in these cells.

## Results

**Microtubule inhibitors restore the AJC in carcinoma cells.** Human colon carcinoma HT29 cells exhibit loose cell–cell association, as judged by a 'halo' along the cell boundaries (Fig. 1a). ZO-1, a TJ protein, was detected as discontinuous puncta (Fig. 1b, upper panel), suggesting that these cells failed to organize normal TJs. Using ZO-1 as a marker, we conducted a high-content screening to search for chemical compounds that can reorganize ZO-1 into the honeycomb-like pattern that is characteristic of normal epithelial cells[23]. Among 160,960 compounds tested, we found 124 compounds to be effective (an example is shown in Fig. 1b, lower panel). Out of these 124 compounds, 48 showed a chemical structure identical or similar to that of known microtubule polymerization inhibitors (MTIs), which include nocodazole (Fig. 1a, Supplementary Data 1). We confirmed that all of these compounds were able to depolymerize microtubules by immunostaining for α-tubulin. Another 55 compounds also exhibited the ability to depolymerize microtubules, although they were not registered as MTIs (Supplementary Table 1). Thus, we estimated 83% of the effective compounds to be microtubule-depolymerizing drugs. On the other hand, microtubule depolymerization inhibitors, such as paclitaxel, did not affect the

junctional morphology of HT29 cells (Fig. 1a). With these results, we decided to investigate how cells respond to MTIs, choosing nocodazole as a representative MTI.

When HT29 cells were treated with 10 μM nocodazole for 60 min (Fig. 1a), a condition that sufficiently diffuses most microtubules, not only ZO-1 but also E-cadherin became concentrated along the apical edges of cell–cell contacts, in contrast with its diffuse distribution along cell–cell boundaries in untreated cells (Fig. 1b–d, HT29). Movies showed that the gaps between cells began to disappear around 20-min incubation with nocodazole (Fig. 1e, Supplementary Movie 1). When nocodazole was removed, cells regained their loose association (Supplementary Fig. 1a). Cytoplasmic proteins known to interact directly or indirectly with E-cadherin, αE- and β-catenin, and other AJC proteins including l-afadin and Par3, were also re-distributed to the apical edges of cell–cell contacts (Fig. 1f), without changing their total amounts (Supplementary Fig. 1b). Thus, AJC-like structures were restored after nocodazole treatment, although their distribution was not completely continuous along cell–cell contacts, showing occasional breaks particularly at tricellular or multicellular junctions (Fig. 1b inset, and 1f arrows).

We then tested how long the nocodazole effects persist. The cells maintained ZO-1 networks for at least 12 h (Supplementary Fig. 1c, top). During the culture, however, cells entered the mitotic phase one by one, becoming rounded, and these cells failed to proceed to further mitotic steps due to the presence of nocodazole (Supplementary Fig. 1c, middle, and Movie 2). When mitosis was inhibited with Mitomycin C, cell layers maintained the adhesive state for at least 3 days (Supplementary Fig. 1c, bottom). Thus, unless cells enter the mitotic cycle, they maintain firm mutual adhesion in the presence of nocodazole.

Next, we tested whether other colon carcinoma lines also respond to nocodazole, choosing Colo205, LS180, and SW620 cells. These cells expressed E-cadherin (Supplementary Fig. 1d), but did not form ZO-1 networks (Fig. 1g). Nocodazole treatment resulted in a significant expansion of ZO-1/E-cadherin-double positive junctions in all cell lines (Fig. 1d, g). When Colo205 cells were cultured in Matrigel, they grew into spheres, the majority of which had no lumen, or if any, multiple lumens. In the presence of nocodazole, however, they often formed a fully polarized cyst with a single lumen (Fig. 1h), suggesting that their ability to establish apico-basal polarity was improved. We also treated Caco2 cells, which are capable of forming the typical AJC despite their carcinoma origin, with 10 μM nocodazole for 60 min, but their AJC organization did not change at least under this treatment condition (Supplementary Fig. 1e). Thus, MTI sensitivity was common to multiple lines of adhesion-defective colon carcinoma cells, but not to the 'well differentiated' carcinoma cells. In mixed cultures of HT29 and Caco2 cells, ZO-1 failed to accumulate at their heterotypic boundaries, indicating that HT29 cell adhesiveness is cell-autonomously defective. However, after nocodazole treatment, ZO-1 became concentrated in these boundaries (Supplementary Fig. 1f), allowing HT29 cells to share junctions with Caco2 cells. We then tested whether there are any differences between HT29 and Caco2 cells with regard to microtubules. Western blots showed no differences in the amount of α-tubulin, or the levels of acetylation or tyrosination of microtubules, between the two cell lines (Supplementary Fig. 1g).

**RhoA and GEF-H1 are required for AJC formation.** We noticed that membrane blebbing occurred on the cell surfaces temporarily soon after nocodazole treatment (Supplementary Movie 1). As blebbing is a hallmark of Rho activation[24, 25], we assayed its activity, and confirmed that active RhoA was increased after

nocodazole treatment (Fig. 2a). We visualized RhoA activation processes using a Raichu probe[26], confirming that the fluorescence labels, which represent higher RhoA activities, increased in a patchy pattern at the apical surfaces and cell–cell contact zones within 10 min (Supplementary Fig. 2a). Immunostaining for RhoA in fixed specimens also showed that RhoA was distributed diffusely in control HT29 cells, whereas it became intensified at

apical zones of the cytoplasm, including cell junctions, after nocodazole treatment (Fig. 2b, c), even though its total expression level did not change (Supplementary Fig. 2b). To test whether RhoA is involved in AJC induction, we depleted RhoA using siRNAs (Supplementary Fig. 2c), and found that RhoA depletion suppressed the junction reorganization (Fig. 2d). One of the RhoA effectors, ROCK1, was also relocalized to apical cytoplasm

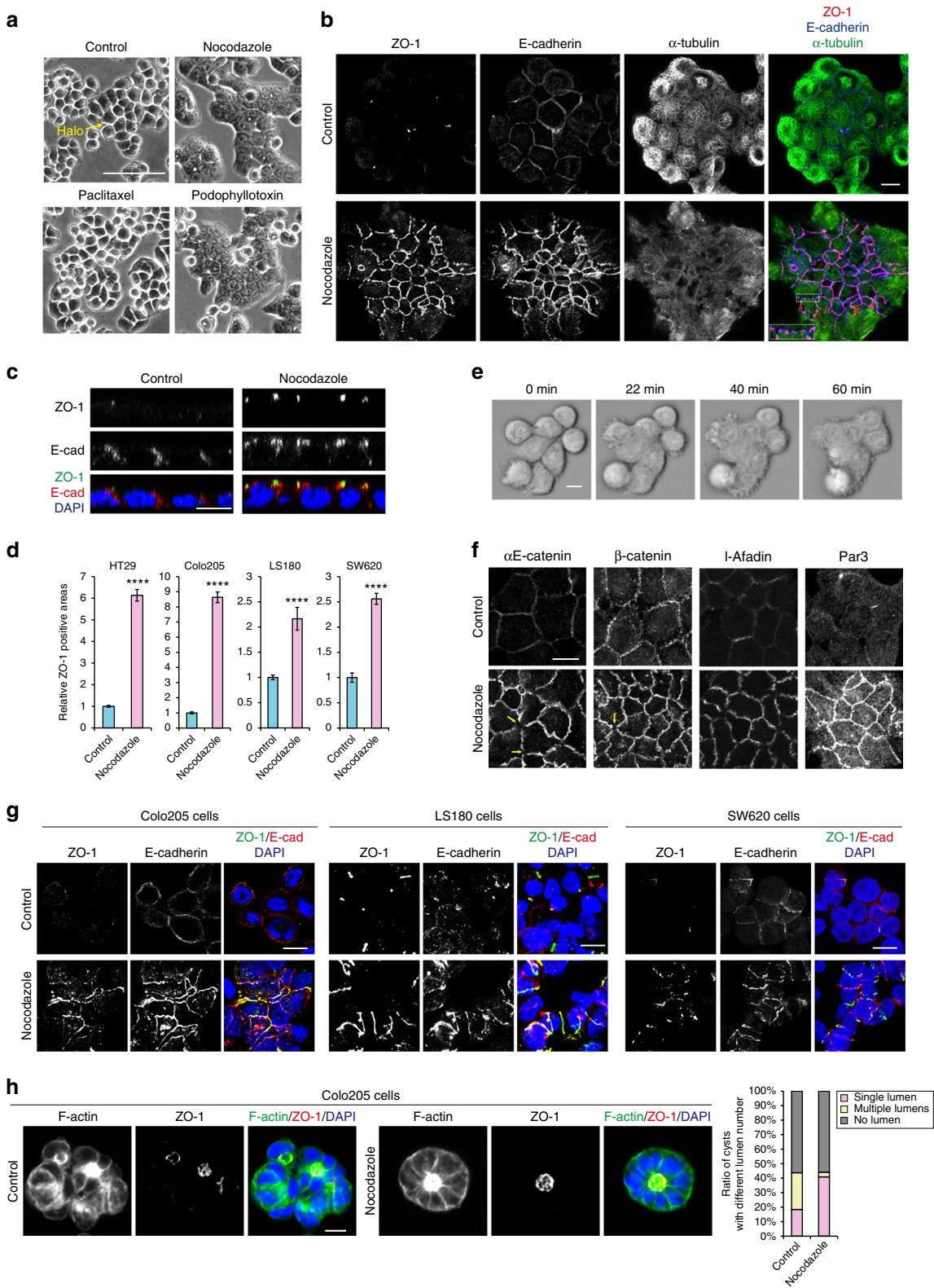

**Fig. 1** Microtubule depolymerization induces the apical junctional complex in colon carcinoma cells. **a** Phase-contrast images of HT29 cells. Cells were treated with 10 μM paclitaxel, 10 μM nocodazole, or 10 μM podophyllotoxin in 0.1% DMSO for 1 h. Control cells were treated only with 0.1% DMSO throughout the experiments. **b** Immunostaining for ZO-1, E-cadherin and α-tubulin at the apical plane of HT29 cells treated with 10 μM nocodazole for 1 h. This condition for nocodazole treatment of HT29 cells was used throughout the experiments unless otherwise noted. Boxed area is an example of discontinuous ZO-1/E-cadherin distributions. **c** Lateral views of HT29 cells stained for ZO-1, E-cadherin and DNA (with DAPI). **d** Relative ZO-1 positive areas. See Methods section for details of quantification. $n = 86$ fields for both control and nocodazole-treated HT29 cultures, pooled from three independent experiments. $n = 61$ for both control and nocodazole-treated Colo205 cultures, $n = 68$ for both control and nocodazole-treated LS180 cultures, and $n = 67$ and 64 for control and nocodazole-treated SW620 cultures, respectively, pooled from two independent experiments. Mean ± SEM. ****$P < 0.0001$ by two-tailed Mann–Whitney U-test. **e** Time-lapse images of HT29 cells treated with 20 μM nocodazole. Images are identical to Supplementary Movie 1. **f** Immunostaining for αE-catenin, β-catenin, l-afadin, and Par3 in HT29 cells. Arrows indicate examples of junctional areas that lack catenins. **g** Staining for ZO-1, E-cadherin and DNA in Colo205, LS180, and SW620 cells treated with nocodazole for 1 h. Relative ZO-1 positive areas are shown in **d**. **h** Colo205 cysts formed in Matrigel, stained for F-actin, ZO-1 and DNA. The cells were cultured for 3 days, and then treated with 0.2% DMSO or 20 μM nocodazole in 0.2% DMSO for 3 h. The ratio of cysts with a single, multiple and no lumens is shown. $n = 87$ cysts for control and 93 cysts for nocodazole treatment, pooled from two independent experiments. Scale bars, 100 μm **a**; 10 μm **b–h**. See also Supplementary Fig. 1

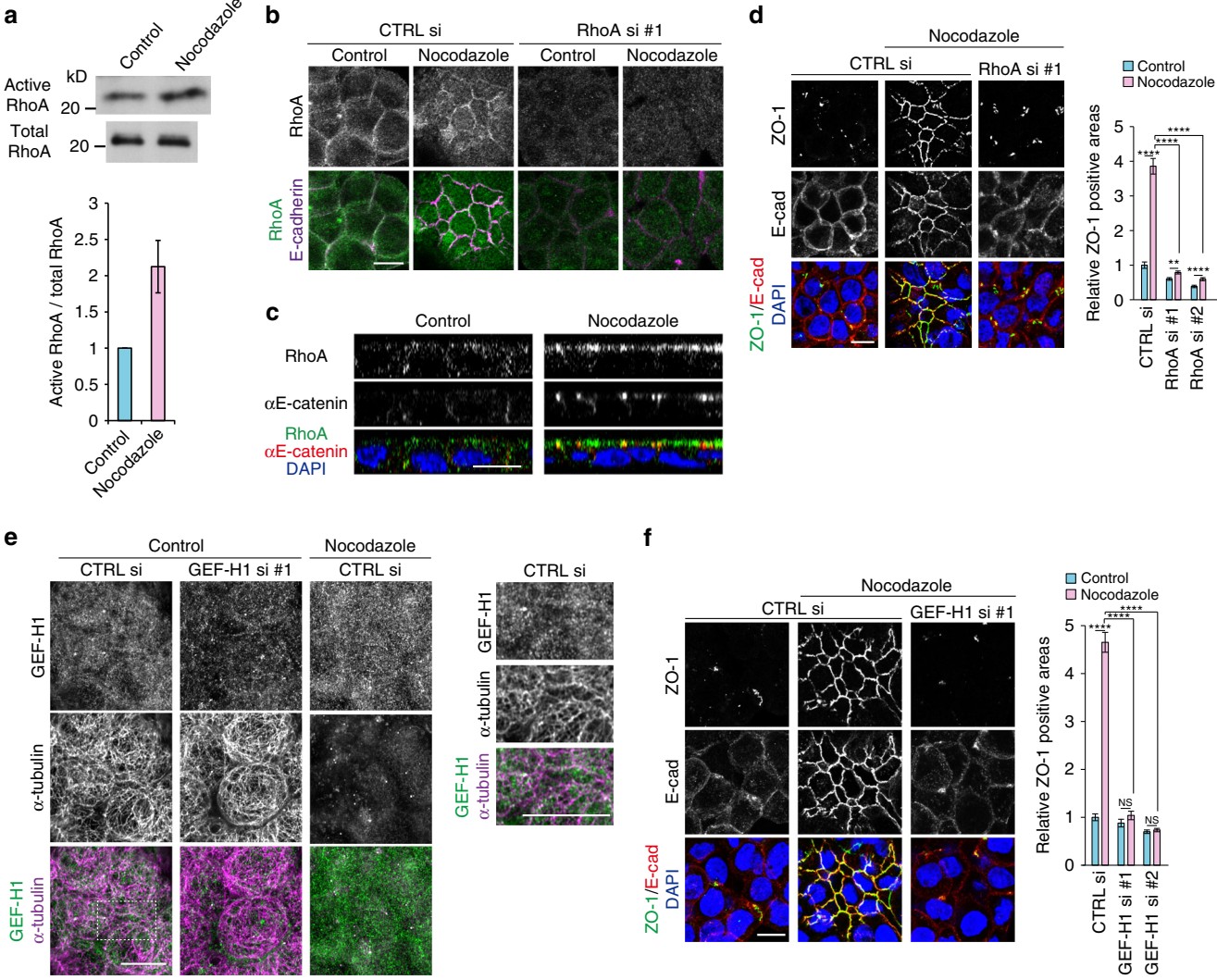

**Fig. 2** RhoA upregulation is essential for apical junction formation. **a** Assay of RhoA activity using the lysates of HT29 cells treated or not treated with nocodazole. Active and total RhoA were detected by immunoblotting for anti-RhoA antibody. The graph shows the ratio of active to total RhoA, assessed by densitometric scanning of blots. $n = 5$ independent experiments, mean ± SEM. **b** Immunostaining for RhoA and E-cadherin in HT29 cells transfected with control (CTRL) or RhoA siRNA (si). **c** Lateral views of HT29 cells immunostained as in **b**. **d** Staining for ZO-1, E-cadherin and DNA in HT29 cells transfected with CTRL si or RhoA si. Relative ZO-1 positive areas are also shown. $n = 65, 65, 71, 70, 66$ and 71 fields for nocodazole-untreated (control) and -treated cells, which were transfected with CTRL si, RhoA si #1 and RhoA si #2, respectively, pooled from two independent experiments. **$P = 0.0011$, ****$P < 0.0001$ by two-tailed Mann–Whitney U-test. **e** Immunostaining for GEF-H1 and α-tubulin in HT29 cells transfected with GEF-H1 siRNA. Images were obtained with Airyscan. Part of the control image is enlarged at the right. **f** Staining for ZO-1, E-cadherin and DNA in HT29 cells transfected with CTRL or GEF-H1 siRNAs. Relative ZO-1 positive areas are also shown. $n = 69, 67, 68, 68, 69,$ and 72 fields for nocodazole-untreated and -treated cells, which were transfected with CTRL si, GEF-H1 si #1 or GEF-H1 si #2, respectively, pooled from two independent experiments. ****$P < 0.0001$ by two-tailed Mann–Whitney U-test. NS, not significant. Scale bars, 10 μm. See also Supplementary Fig. 2

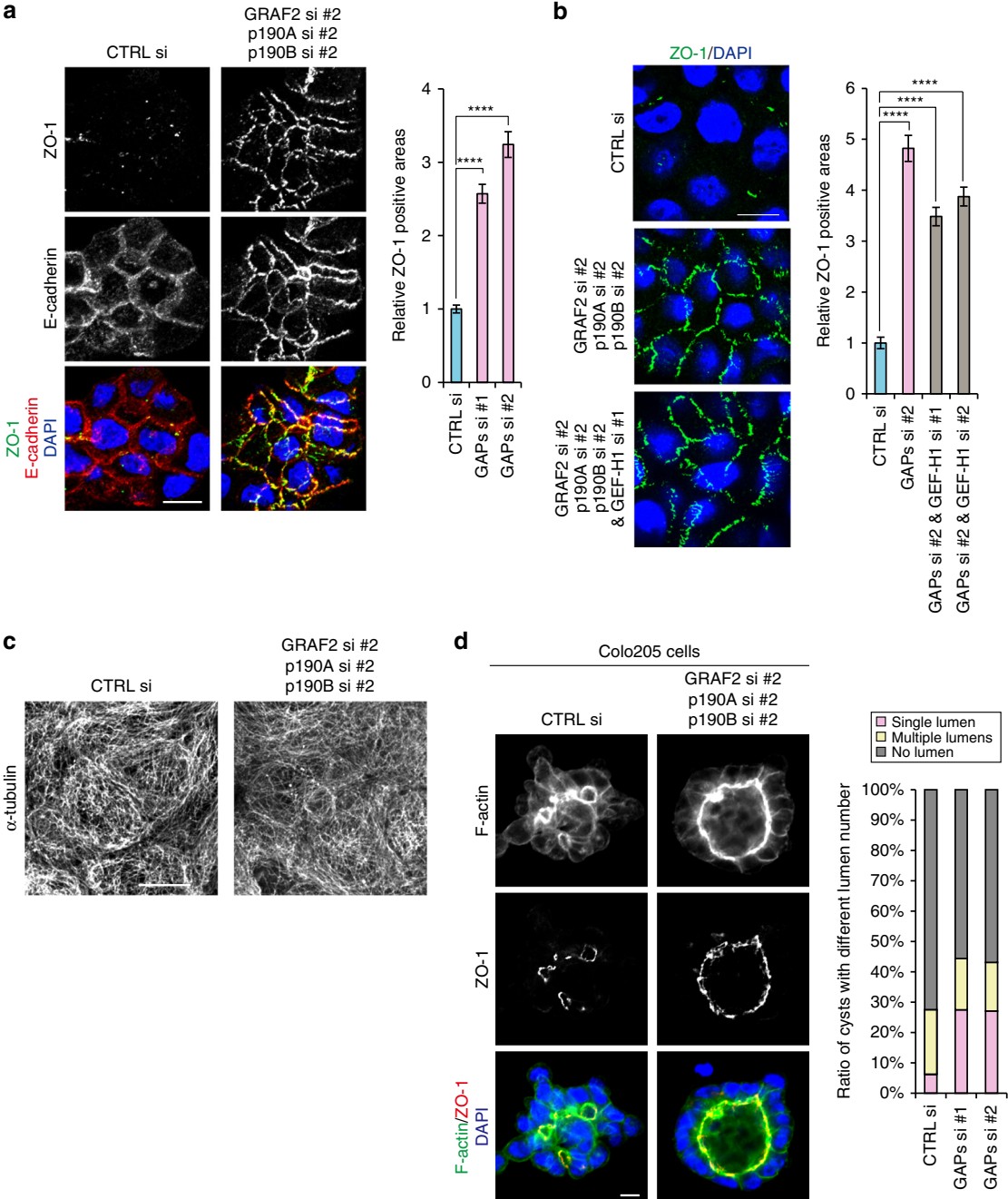

**Fig. 3** RhoGAPs depletion induces junction recovery. **a** Staining for ZO-1, E-cadherin and DNA in HT29 cells transfected with CTRL si or a mix of GRAF2 si #2, p190A si #2 and p190B si #2 (GAPs si #2). Relative ZO-1 positive areas are also shown. $n = 77$, 72, and 69 fields for cells transfected with CTRL si, GAPs si #1 (a mix of GRAF2 si #1, p190A si #1 and p190B si #1) and GAPs si #2, respectively, pooled from two independent experiments. ****$P < 0.0001$ by two-tailed Mann–Whitney $U$-test. **b** Staining for ZO-1 and DNA in HT29 cells transfected with CTRL siRNA or GAPs si #2 or a mix of GAPs si #2 and GEF-H1 si #1. Relative ZO-1 positive areas are also shown. $n = 45$, 43, 45, and 42 fields for cells transfected with CTRL si, GAPs si #2, a mix of GAPs si #2 and GEF-H1 si #1 or a mix of GAPs si #2 and GEF-H1 si #2, respectively, pooled from two independent experiments. ****$P < 0.0001$ by two-tailed Mann–Whitney $U$-test. **c** Immunostaining for α-tubulin in HT29 cells transfected as in **a**. Images were obtained with Airyscan. **d** Colo205 cysts transfected with CTRL or GAPs si, which were stained for F-actin, ZO-1 and DNA. Cells were cultured in Matrigel for 3 days. The ratio of cysts with a single, multiple and no lumens is shown. $n = 207$ cysts for CTRL si, 178 and 181 cysts for GAPs si #1 and #2, respectively, pooled from two independent experiments. See also Supplementary Fig. 3

as well as to cell junctions after nocodazole treatment without changing its total amount (Supplementary Fig. 2d, e), and the ROCK inhibitor Y-27632 suppressed AJC reformation (Supplementary Fig. 2f). These findings suggest that nocodazole upregulated the activity of the RhoA–ROCK pathway and this upregulation was required for AJC rearrangement.

Next, we explored how MTIs upregulated RhoA. As a potential candidate for RhoA activator, we focused on GEF-H1, since its activity is inhibited by its binding to microtubules[27]. In untreated HT29 cells, GEF-H1 broadly localized in the cytoplasm, partly co-localizing with microtubules (Fig. 2e). After nocodazole treatment, the soluble fraction of GEF-H1 increased, as expected

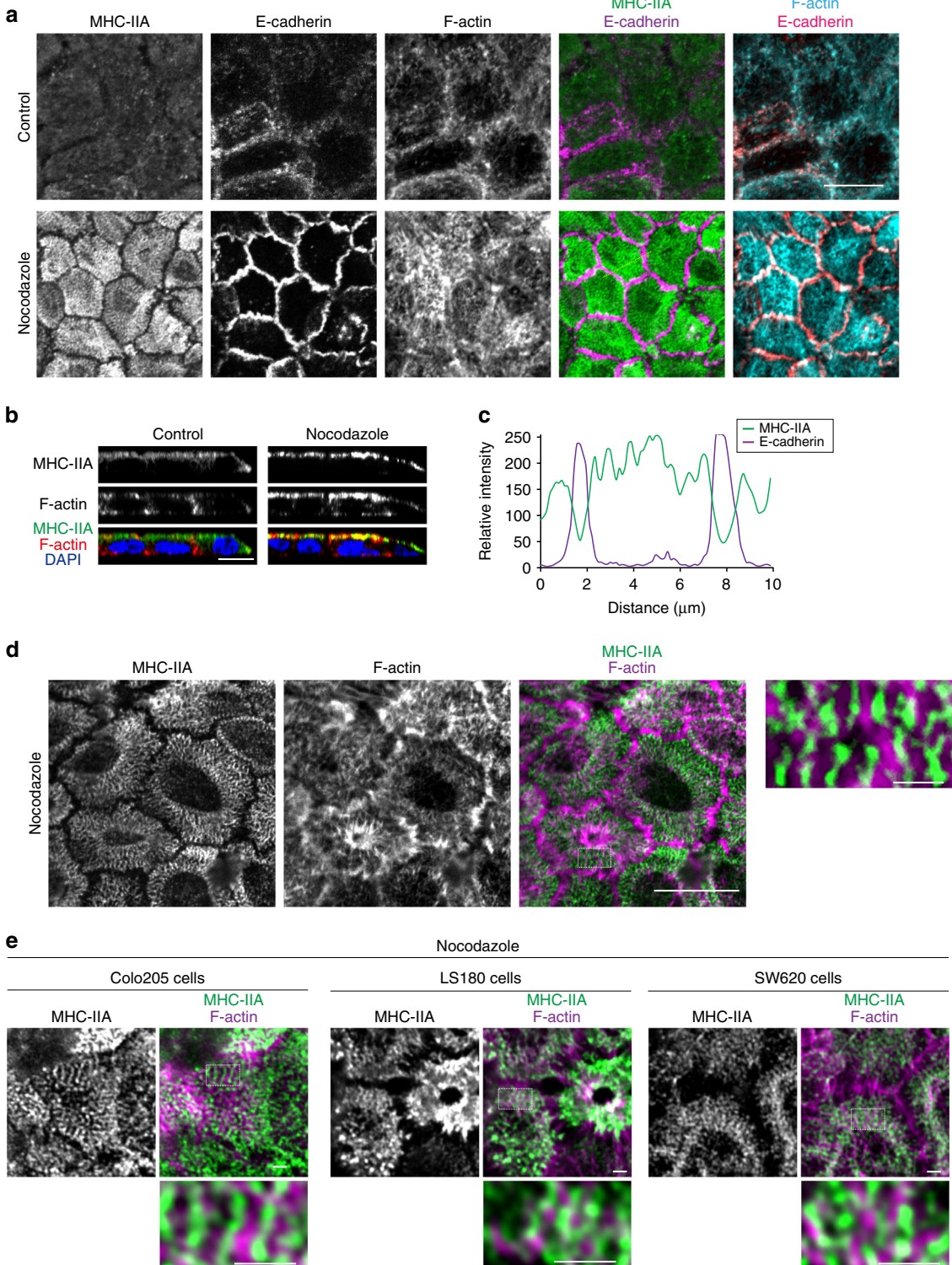

**Fig. 4** Reorganization of cortical myosin-IIA by nocodazole treatment. **a** Immunostaining for MHC-IIA, E-cadherin and F-actin in HT29 cells. **b** Lateral views of cells immunostained as in **a**. **c** Densitometric scans of immunosignals for MHC-IIA and E-cadherin along the yellow line drawn in **a**. **d**, **e** Airyscan images for MHC-IIA and F-actin at the apical plane of HT29 (**d**), and Colo205, LS180 and SW620 (**e**) cells, treated with nocodazole. Box regions are enlarged. Scale bars, 10 μm **a**, **b**, **d**; 1 μm **d** right, **e**. See also Supplementary Fig. 4

(Supplementary Fig. 2g), but its overall distribution did not particularly change, showing no concentration at cell junctions. Then, we depleted GEF-H1 using siRNAs (Supplementary Fig. 2h), and found that GEF-H1 loss abolished nocodazole-dependent AJC formation (Fig. 2f), indicating that GEF-H1 is involved in this process.

**Depletion of RhoGAPs is sufficient to induce AJC formation.** To confirm that RhoA activation was critical for AJC formation, we tested the effects of depletion of RhoGAPs that inactivate RhoA[28]. HT29 cells expressed at least three GAPs—GRAF2, p190A and p190B—therefore we co-depleted them (Supplementary Fig. 3a). The results showed that co-depletion of these

molecules induced formation of AJCs whose morphology is indistinguishable from that in nocodazole-treated cells (Fig. 3a), supporting the idea that Rho activity is crucial for this process. Immunostaining for RhoA and ROCK1 also showed that both of them increased at apical zones of these cells (Supplementary

Fig. 3b). Importantly, even when GEF-H1 was removed, RhoGAPs-depleted cells still acquired AJCs (Fig. 3b), which indicates that GEF-H1 is no longer required once RhoA was activated via GEF-H1- independent routes. Furthermore, in these RhoGAPs-depleted cells, microtubule networks appeared

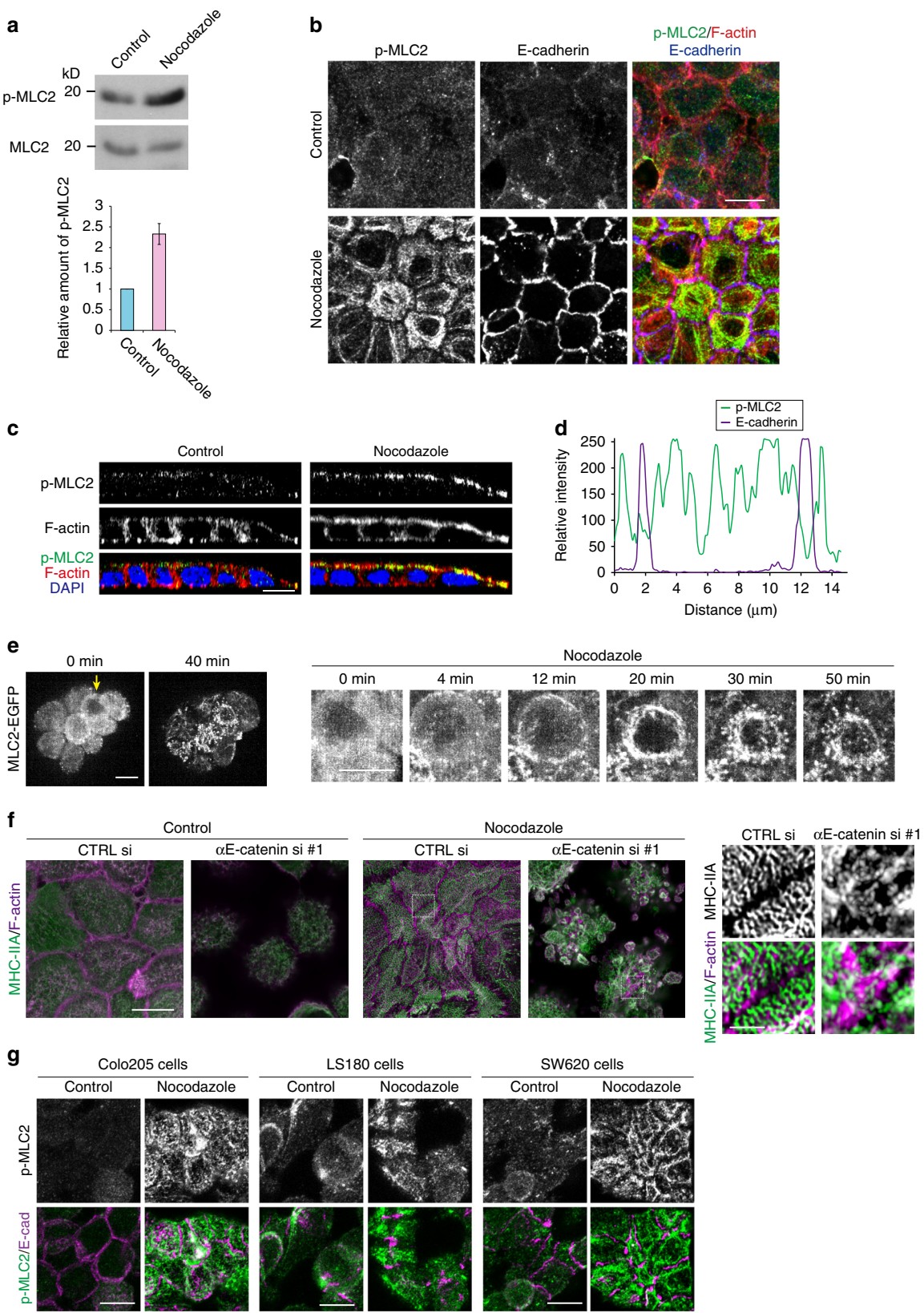

comparable to those in control cells (Fig. 3c), indicating that microtubules are not inhibiting junction formation. These suggest that a crucial process for junctional reorganization in HT29 cells is to gain higher RhoA activities, but not to lose microtubules. Therefore, the MTI-mediated GEF-H1 upregulation can be regarded as merely one of the ways to activate RhoA. We also confirmed that RhoGAPs depletion in Colo205 cells promoted formation of cysts with a single lumen in Matrigel cultures (Fig. 3d).

**Reorganization of cortical myosin II**. One of the targets for the Rho/ROCK system is non-muscle myosin II. Strikingly, after nocodazole treatment, myosin-IIA became densely condensed at the apical cortex, except at its central regions (Fig. 4a, b), without changing its total amount (Supplementary Fig. 4a). Simultaneously, the density of apical F-actin also increased (Fig. 4b). Depletion of myosin-IIA heavy chain (MHC-IIA) severely inhibited the nocodazole-mediated AJC formation (Supplementary Fig. 4b, c), confirming its importance in AJ reorganization. Curiously, the rearranged myosin-IIA did not overlap with junctional E-cadherin (Fig. 4c), suggesting that its redistribution was purely a cortical event. On the other hand, F-actin distributed across the cortical and junctional areas (Fig. 4a).

Super-resolution microscopy showed that MHC-IIA, detected with antibodies that recognize the tail region, were arranged in a sarcomere-like striated pattern (Fig. 4d), as observed for myosin-II 'stacks' that were defined as clusters of myosin II filaments, which periodically localize along actin stress fibers[29]. F-actin was often intercalated between MHC-IIA stacks (Fig. 4d, right), but the entire spatial relation between F-actin and MHC-IIA was not well ordered, contrasted with the case of stress fibers. Similar reorganization of actomyosin was also observed in other carcinoma lines (Fig. 4e), but not in Caco2 cells.

Myosin II is activated by phosphorylation of regulatory light chain (MLC2), and therefore we analyzed the phosphorylated form of MLC2 (p-MLC2). Western blots showed that p-MLC2 was increased after nocodazole treatment (Fig. 5a). Immunostaining showed that p-MLC2 was diffuse in control cells, whereas after nocodazole treatment, it became strongly condensed at apical cortical regions in a ring-like pattern, again avoiding E-cadherin (Fig. 5b–d). Co-immunostaining for p-MLC2 and the MHC-IIA tail showed that p-MLC2 was irregularly associated with MHC-IIA stacks (Supplementary Fig. 5a). The apical condensation of p-MLC2 was inhibited with ROCK or myosin inhibitors, or by GEF-H1 depletion (Supplementary Fig. 5b, c), confirming that GEF-H1 and ROCK work upstream of MLC2. We also confirmed that MLC2 was activated by RhoGAPs depletion (Supplementary Fig. 5d).

To explore the dynamics of myosin-II reorganization, we transfected cells with MLC2-EGFP and acquired live images. Within 10 min after nocodazole administration, the cortical networks of MLC2 began to shrink (Fig. 5e, Supplementary Movie 3). In cells surrounded by other cells, the MLC2 networks centripetally contracted forming a ring, whereas those located at the periphery of colonies displayed more irregular condensation,

suggesting that the mode of contraction is affected by cell–cell contacts. To test this idea, we observed cells from which αE-catenin or E-cadherin was depleted (Supplementary Fig. 5e), and found that myosin-IIA networks still became condensed in these cells, but without forming stripes (Fig. 5f, Supplementary Fig. 5f, g). This suggests that the sarcomere-like reorganization of MHC-IIA depends on CCC-mediated cell–cell adhesion.

Nocodazole-sensitive cortical p-MLC2 reassembly also occurred in Colo205, LS180, and SW620 cells (Fig. 5g). We also checked whether p-MLC2 in caco2 cells similarly responds to nocodazole treatment. Western blot and immunostaining analysis showed that, in caco2 cells, p-MLC2 tended to be slightly upregulated after nocodazole administration, but not as drastically as in HT29 cells (Supplementary Fig. 5f, h, i), verifying that only AJC-defective carcinoma cells are exceptionally sensitive to microtubule depolymerization.

**F-actin links myosin-II networks and cell junctions**. To investigate how myosin-IIA controls AJC formation without localizing to this structure, we explored the possibility that F-actin may link the cortex and junction, as it distributes across the two structures. To observe actin behavior in detail, we took live images of Lifeact-EGFP introduced into HT29 cells. Before nocodazole treatment, the borders of F-actin–labeled cells were smooth in appearance. After the treatment, however, the cell borders became converted into filamentous structures, which are expediently called 'protrusions' here (Fig. 6a, top, Supplementary Movie 4), with retraction of the overall cell peripheries. Such protrusions, on the other hand, did not appear in αE-catenin–depleted cells, although their surfaces vigorously blebbed (Fig. 6a, bottom, Supplementary Movie 5). These suggest that the protrusions were formed as a structure that links the junctional CCC and the cortical acto-myosin networks that are actively contracting. If so, these protrusions should be under strain, and we tested this idea by a laser ablation experiment. When protrusions were cut by a laser, both ends of the cut surface retracted, but more vigorously toward the cortex side (Fig. 6b, Supplementary Movie 6), implying that the pulling force primarily originates from the cortex. These results suggest that cortical actomyosin exerts tension on cell junctions via F-actin.

**Vinculin recruitment to cell junctions**. To study how the CCC responds to tension, we searched for molecules that are recruited to cell junctions in a myosin-II–dependent way, and found that vinculin is one such molecule. Unlike most junctional proteins that are detected on the cell surface even in untreated HT29 cells (Fig. 1f), vinculin accumulated at cell–cell contacts only when cells were treated with nocodazole (Fig. 6c). This junctional accumulation of vinculin was blocked with Y-27632 or blebbistatin (Supplementary Fig. 6a). These observations are consistent with the previous finding that vinculin is recruited to cell junctions by binding to αE-catenin only when this catenin is tugged by actomyosin force[30, 31].

To analyze the role of vinculin, we depleted it with specific siRNAs (Supplementary Fig. 6b), and found that junctional αE-

**Fig. 5** Myosin-II dynamics in nocodazole-treated cells. **a** Upregulation of phosphorylated MLC2 (p-MLC2) after nocodazole treatment, assessed by western blotting. Densitometric comparison for the electrophoretic bands is also shown. $n = 4$ independent experiments, mean ± SEM. **b** Immunostaining for p-MLC2, F-actin, and E-cadherin in HT29 cells. **c** Lateral views of HT29 cells immunostained as in **b**. **d** Densitometric scans of immunosignals for p-MLC2 and E-cadherin along the yellow line drawn in **b**. **e** Time-lapse images of MLC2-EGFP expressed in HT29 cells during nocodazole treatment. Images at the right are a close-up view of the cell indicated by arrow at the left panel. Images are identical to Supplementary Movie 3. **f** Effects of αE-catenin knockdown on MHC-IIA distribution, as assessed by immunostaining of HT29 cells transfected with control or αE-catenin specific siRNA. Images were obtained with Airyscan. **g** Effects of nocodazole treatment on the distribution of p-MLC2 and E-cadherin in Colo205, LS180, and SW620 cells. Scale bars, 10 μm **b**, **c**, **f** left and **g**; 2 μm (**f** right). See also Supplementary Fig. 5

catenin or E-cadherin became fragmented, and each fragment was oriented perpendicularly to the axis of cell–cell contacts, co-localizing with actin filaments (Fig. 6c, d). This implies that these filaments radially pull CCC components when vinculin is lost. To confirm this pulling force, we prepared Lifeact-EGFP-labeled cells in which vinculin was depleted, and applied laser ablation to the filamentous junctional structures. The results

showed that their retraction was induced by a pattern similar to that found in the peripheral 'protrusions' of nocodazole-treated control cells (Supplementary Fig. 6c, Supplementary Movie 7). These observations suggest that vinculin works for making the CCC-based junctions resistant to radial force. In these cells, ZO-1 networks were also disrupted (Supplementary Fig. 6d).

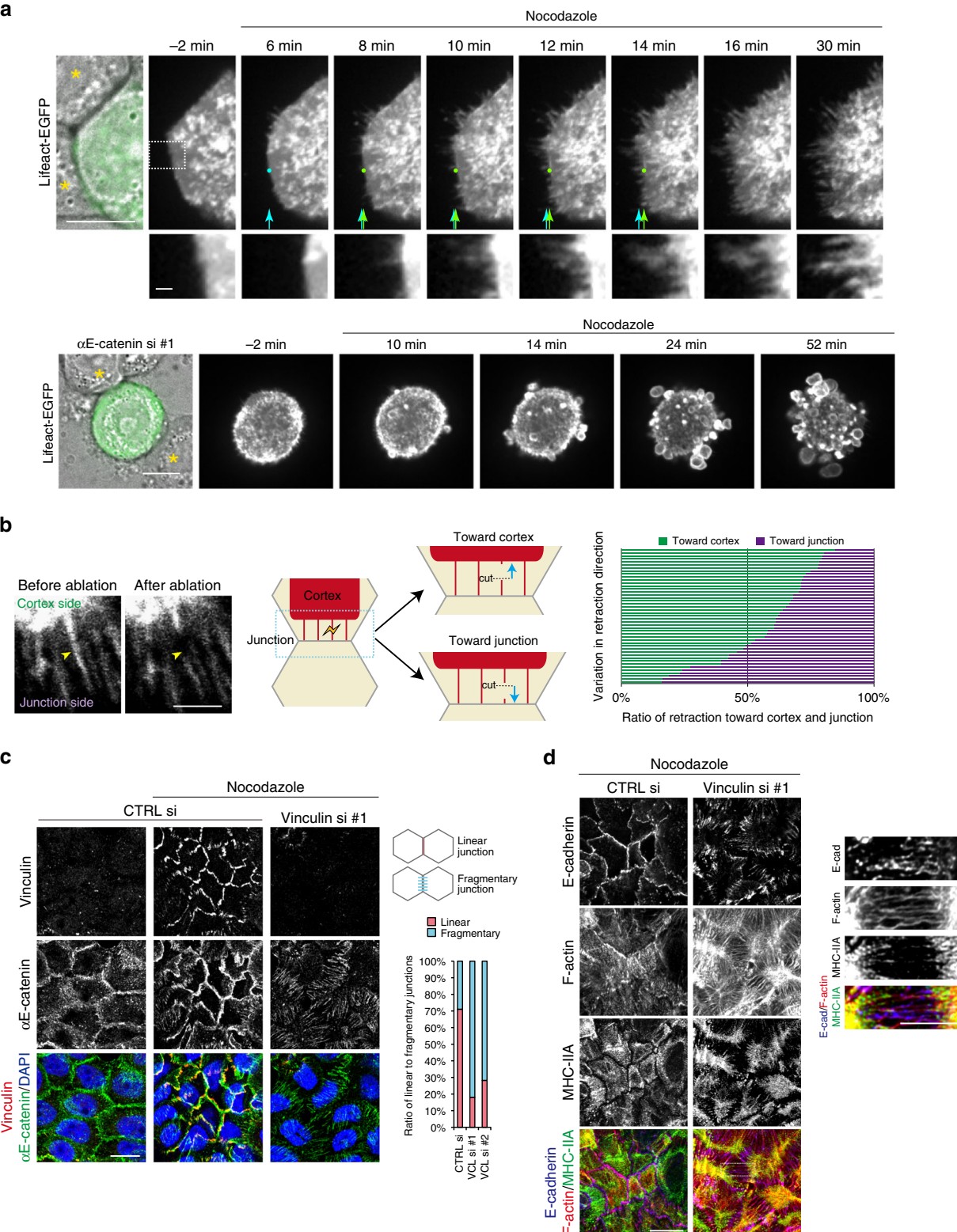

**Vinculin strengthens junctions by recruiting PDZ-RhoGEF.**
How does vinculin reinforce junctions? Nocodazole upregulated
RhoA and ROCK not only at cortices but also along cell junctions, despite no particular concentration of GEF-H1 at the
junctions (Fig. 2e), suggesting that other factors may control the
junctional RhoA. In fact, we found that another RhoGEF, PDZ-
RhoGEF, was recruited to junctions after nocodazole treatment,
and this RhoGEF co-localized with RhoA or ROCK (Fig. 7a,
Supplementary Fig. 7a). Importantly, this PDZ-RhoGEF re-
localization was vinculin-dependent, but not vice versa, as
assessed by analyzing the cells from which vinculin or PDZ-
RhoGEF was depleted (Fig. 7b, Supplementary Fig. 7b), indicating
that vinculin plays a role in recruiting PDZ-RhoGEF to junctions.
Consistently, junctional PDZ-RhoGEF co-localized with vinculin
(Fig. 7b). Then, we examined the effects of PDZ-RhoGEF
depletion, and found that it perturbed the nocodazole-induced
AJC formation. Although E-cadherin became condensed to apical
regions, it became radially oriented, and radial actin filaments
anchored to it (Fig. 7c, Supplementary Fig. 7c), as seen in
vinculin-depleted cells. This supports the idea that PDZ-RhoGEF
works downstream of vinculin. On the other hand, PDZ-RhoGEF
depletion did not inhibit cortical MHC-IIA or p-MLC2 condensation (Fig. 7c, Supplementary Fig. 7d), implying that PDZ-
RhoGEF specifically controls cell junctions, but not cortices.

How does PDZ-RhoGEF control the junctions? Although its
effectors Rho and ROCK colocalize with PDZ-RhoGEF, their
well-known target myosin-II is absent at junctions. We tested the
possibility that the PDZ-RhoGEF/Rho/ROCK signaling may
control actin polymerization, as ROCK is known to target the
LIM kinase (LIMK)–cofilin pathway, in which ROCK activates
LIMK, and the activated LIMK phosphorylates cofilin at Ser-3,
resulting in inhibition of the function of cofilin that depolymerizes actin filaments[32–34]. To test potential involvement of
cofilin, we prepared HT29 cells stably expressing a
phosphorylation-mimic S3E-cofilin or dephoshorylation-mimic
S3A-cofilin[34], and found that S3A-, but not S3E-, cofilin
expression interfered with nocodazole-mediated junction reorganization (Supplementary Fig. 7e), indicating that cofilin needs to
be inactivated for junction formation. Then, we depleted LIMK1
in HT29 cells (Supplementary Fig. 7b) and found that E-cadherin
and actin filaments were altered in a way similar to that observed
in PDZ-RhoGEF depleted cells (Fig. 7d), that is, cortical myosin-
IIA remains condensed but actin filaments were stretched across
cell junctions. Moreover, co-depletion of cofilin rescued, at least
in part, the defects observed after LIMK1 or PDZ-RhoGEF
depletion (Fig. 7d, e, Supplementary Fig. 7b). All these results
support the idea that the PDZ-RhoGEF/Rho/ROCK signaling
inhibits cofilin activity by phosphorylating LIMK, thus leading to
F-actin stabilization specifically at cell–cell junctions.

To further confirm that suppression of cofilin activity
contributes to the junctional reinforcement by vinculin, we
co-depleted cofilin and vinculin in HT29 cells. Cofilin depletion
considerably suppressed the radial re-orientation of CCC

components induced by vinculin depletion, although not perfectly
(Fig. 7f, Supplementary Fig. 7f), consistent with the idea that the
vinculin-dependent recruitment of the PDZ-RhoGEF/LIMK/
cofilin system to cell–cell contact sites plays a role in strengthening cell junctions via F-actin stabilization. Incomplete rescue by
cofilin depletion, however, suggests that other factors may also be
involved in this process.

**Mechanical changes of nocodazole-treated cells.** Nocodazole-
induced cortical changes likely altered mechanical properties of
the cell surface. Therefore, we measured the stiffness of the surface of HT29 cells with atomic force microscopy (AFM), by
pushing cells with a glass bead attached to a cantilever in an
approximately 6-µm depth (Fig. 8a, b). Stiffness of the cell surface, thus measured, dramatically increased after nocodazole
treatment (Fig. 8c). Such increase was also observed in αE-catenin
depleted cells, but to a much lesser extent. These observations
indicate that cortical actomyosin contraction generally increases
the surface stiffness, but this effect is strengthened by AJC formation. Thus, we confirmed that AJC formation is important for
providing cells with proper stiffness.

**Discussion**
Our findings indicate that the following molecular events rebuild
AJCs in adhesion-defective colon carcinoma cells: microtubule
disruption activated GEF-H1, and in turn upregulated RhoA. This
process then induced the contraction of cortical actomyosin
networks. Actin filaments of these networks exerted radial tension
to the CCC, resulting in vinculin recruitment to the junctions.
This process further recruited PDZ-RhoGEF to the junctions,
which promoted actin assembly through a control of RhoA/
LIMK/cofilin signaling (Fig. 9). Importantly, the initial MTI-
treatment step could have been bypassed by removing RhoGAPs,
and, in this case, GEF-H1 was not required for junctional reorganization. This suggests that RhoA upregulation is the key to
trigger the AJC reorganizing cascade, regardless of how RhoA is
activated; the MTI-mediated GEF-H1 activation must have
operated only as a way to activate RhoA. Our results also suggest
that two pools of RhoA were involved in junction recovery. As
seen in nocodazole-treated cells, RhoA was initially activated by
GEF-H1 at the cortex, and then by PDZ-RhoGEF at the junctions
in response to GEF-H1–triggered cortical tension (Fig. 9b).
RhoGAPs-depleted cells mimicked all the processes observed in
nocodazole-treated cells. However, it remains to be investigated
how the cortical and junctional events were coordinated when
RhoA was globally upregulated in the cytoplasm by RhoGAPs
depletion.

Early studies showed that Colo205 cells, which are normally
dispersed, were reorganized into epithelial sheets in response to
various treatments, one of which was to over-express cadherin
mutants that are unable to bind p120-catenin[19, 20]. We suspect
that this process could also involve RhoA activation. p120-catenin

**Fig. 6** F-actin dynamics and vinculin recruitment at cell junctions. **a** Time-lapse images of Lifeact-EGFP expressed in HT29 cells during nocodazole
treatment in control (upper panels) or αE-catenin depleted cells (lower panels). Protrusions emerged from the cell periphery in control cells, whereas only
bubbling occurred in the absence of αE-catenin. Light blue arrows/dot indicate the original position of Lifeact-EGFP positive edges, and this edge is shifted
inward as indicated by light green arrows/dots. Asterisks label neighboring cells. Boxed area is enlarged below. Images are identical to Supplementary
Movies 4, 5. **b** Laser ablation of Lifeact-EGFP labeled protrusions in HT29 cells treated with nocodazole. Yellow arrowheads indicate the site for laser
application. Length of retraction toward the cortical and junctional sides was measured, and the ratio of the two values is shown. $n = 45$ protrusions pooled
from three independent experiments. Images are identical to Supplementary Movie 6. **c** Effects of vinculin knockdown on the distribution of αE-catenin. The
ratio of linear to fragmentary junctions was calculated. $n = 4217$ junctions in 91 colonies for CTRL si; and 2026 junctions in 94 colonies and 3239 junctions
in 106 colonies for vinculin si #1 and #2, respectively, pooled from two independent experiments. **d** Effects of vinculin knockdown on the distribution of F-
actin and E-cadherin. Images were obtained with Airyscan. Boxed area is enlarged at the right. Scale bars, 10 µm **a**, **c**, **d**, 5 µm (**d** right) and 1 µm (**a**-enlarged
images, and **b**). See also Supplementary Fig. 5

is known to suppress RhoA[35, 36], and the cadherin-bound p120-catenin prevents actomyosin from contracting at junctional areas[37]. Therefore, removal of p120-catenin from cell membranes by overexpressing p120-catenin uncoupled cadherins might have affected cortical RhoA activity, triggering junctional reorganization according to our model, although this idea needs experimental confirmation.

The most dramatic changes, which occur during adhesion induction, include reorganization of cortical myosin IIA into striped structures. These are likely identical to myosin-II stacks, a cluster of myosin-II filaments, which were previously defined[29]. Myosin-II stacks periodically associate with actin stress fibers in fibroblasts, displaying a sarcomere-like appearance, and they are also detected on actin filaments running along epithelial cell–cell contacts[38, 39]. However, unlike the myosin-II organization in these normal structures, actin filaments and p-MLC in carcinoma cells studied here did not associate in an orderly fashion with MHC-IIA stacks, leaving questions of how the MHC-IIA stacks

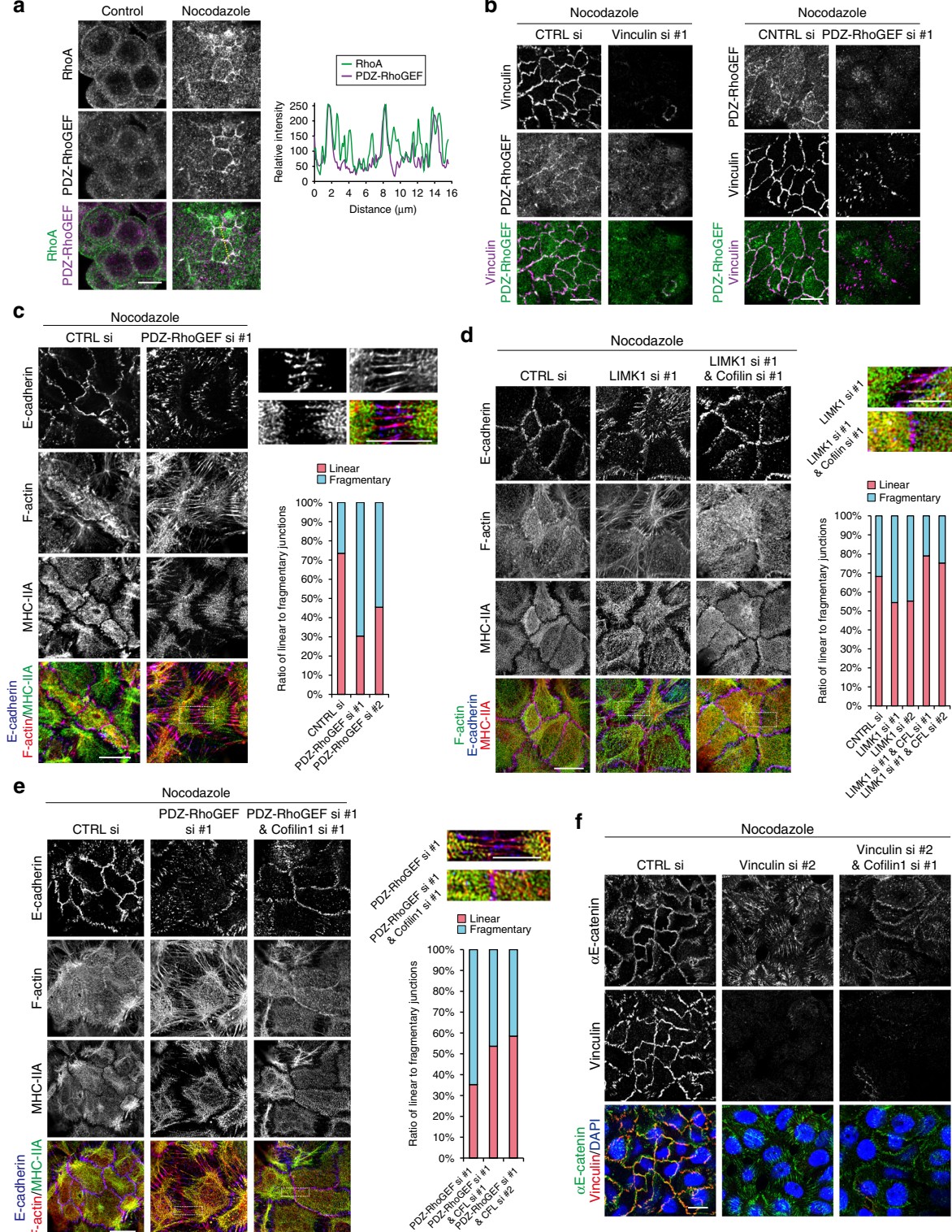

can be arranged in a stripe fashion and how they control the concentric contraction of cortical actomyosin networks. We showed that formation of the periodic pattern of MHC-IIA depended on the CCC, and actin filaments linking the cortex and junction were radially pulled. This suggests that myosin-IIA stacks are rearranged in response to tension, although this putative mechanism needs to be further investigated in detail.

How does the contracting cortical actomyosin reorganize the CCC and other junction proteins? A tugging force is generally exerted to cell–cell junctions[40–42], and the CCC binds to actin filaments more stably under tension than without it[43]. AJC recovery in HT29 cells appeared to have depended on such force. While the CCC and F-actin did not show any organized distribution in untreated HT29 cells, they became co-condensed at apical contact regions after nocodazole treatment, suggesting that the cortical actomyosin-mediated tension strengthened the interaction between the CCC and F-actin. Laser ablation experiments confirmed that cortical F-actin pulled the junctions, possibly via αE-catenin[43]. αE-catenin is known to become 'unfurled' by a tugging force exerted on the C-terminal domain, allowing its binding to vinculin[31, 44–46]. Indeed, vinculin accumulated at cell junctions in correlation with actomyosin contraction. This force-dependent recruitment of vinculin was another key event for rebuilding the AJC, as it secondarily recruited PDZ-RhoGEF to the junctions, although it remains to be investigated how vinculin recruited PDZ-RhoGEF. Concerning molecular players downstream of PDZ-RhoGEF, our results suggest that it regulates RhoA/LIMK/cofilin signaling. However, we cannot exclude the possibility that other mechanisms may also operate downstream of vinculin. For example, vinculin may also recruit Mena/VASP for junction stabilization, as shown earlier[47]. Furthermore, other junctional proteins like afadin, which behave in response to tension[39], could be involved in the force-dependent processes of junction remodeling.

It is of note that multiple colon cancer lines similarly responded to nocodazole treatment, suggesting that their junctions are defective via a common mechanism(s). In contrast, Caco2 cells, another line of colon carcinoma cells that preserves normal epithelial architecture, did not respond to nocodazole. Thus, a subgroup of carcinoma cells gained the property to require extra RhoA signals for junction formation. To elucidate the molecular basis underlying this change in RhoA dependency is an important future subject. Although the cell junctions recovered after nocodazole treatment looked normal, these were not completely normal in two ways. First, they failed to cover all the cell–cell contacts, particularly multicellular junctions. Second, in normal epithelial cells including Caco2 cells, actin filaments run in parallel with the junctional cell membranes except at multicellular junctions[39], whereas, in HT29 cells treated with nocodazole, they perpendicularly terminated at the junctions, although a parallel population of actin filaments may also exist in these cells. Additional approaches are required in order to completely repair abnormal junctions in carcinoma cells.

Apart from the cancer-specific problems, the contraction of cortical actomyosin has been implicated in cell junction remodeling during normal morphogenesis using invertebrate embryos[48]. For example, in Drosophila embryos, a pulsating contraction of cortical actomyosin networks contributes to the apical constriction of ventral furrow cells via their linkage to junctional F-actin[49, 50]. As for vertebrate cells, however, only limited examples of the functional linkage between cortical actomyosin and junctions have been reported in normal cells[51]. Whether the cortical tension–mediated remodeling of cell junctions uncovered in the present study is specific to cancer cells or applicable to normal vertebrate cells remains to be determined.

In addition to junctional reorganization, the stiffness of HT29 cells dramatically increased after nocodazole treatment. Metastatic cancer cells are thought to be 'softer' than normal cells, enabling them to pass through structural barriers[52–54]. This suggests that the surface stiffness of cells is controlled by cortical actomyosin contraction and junction formation, offering insight into why invasive cancer cells are soft. Further analysis of MTI/RhoA-dependence of cancer cell adhesion may provide clues to the future development of chemotherapy to suppress tumor dispersion.

## Methods

**High content screening.** Compound libraries were provided by drug discovery initiative (DDI), the University of Tokyo (141,120 compounds), and RIKEN NPDepo library (19,840 compounds). HT29 cells stably expressing EGFP-ZO-1 were prepared and maintained in DMEM/Ham's F12 supplemented with 10% FBS and 1% penicillin-streptomycin. Cells ($2.4 \times 10^4$ cells in 40-μl medium per well) were seeded into wells of CellCarrier-384 TC plates (PerkinElmer cat#600007558) using a multi-channel-pipet, and incubated for 16–18 h at 37 °C in 5% $CO_2$. Chemical compounds were added to the plates using an automatic pipette multi-dispenser (Biotec EDR-384). Final concentrations of DDI library and NPDepo library compounds (see below) were 10 μM and 1 μg/ml, respectively. The plates were incubated for 5–6 h at 37 °C in 5% $CO_2$. The cells were fixed with 4% PFA at room temperature for 30 min. In order to capture cell images and fluorescence signals of EGFP-ZO-1, we used Opera (PerkinElmer OperaQEHS) and the Opera-TwisterII-iLinkPro system (PerkinElmer TwisterII cat#79838, iLinkPro Ver.1.03). Images were acquired by Opera using a ×40 water immersion objective with the following channels: Ex1Cam1 using 488-nm laser excitation and 520/35-nm emission filters for fluorescence signal of EGFP-ZO-1, and Ex2Cam3 using 690-nm TLED and 690/70-nm emission filters for bright fields. The images were analyzed using Columbus (PerkinElmer). The ratio of tight junction area (identified by linear intense fluorescence signal of EGFP-ZO-1) to the total cell area was calculated for each well.

**Identification of microtubule-depolymerizing compounds.** Identification of microtubule depolymerization drugs was performed using HeLa cells. α-Tubulin immunofluorescence staining of HeLa cells was carried out as follows. The cells were rinsed in phosphate buffered saline (PBS) at 37 °C and fixed in methanol + 1 mM EGTA for 3 min at −30 °C. The cells were permeabilized with

**Fig. 7** PDZ-RhoGEF controls junctional F-actin via LIMK and cofilin. **a** Immunostaining for PDZ-RhoGEF and RhoA in HT29 cells. Relative intensity of PDZ-RhoGEF and RhoA signals were scanned along the yellow broken line. **b** Effects of vinculin knockdown on PDZ-RhoGEF (left) or PDZ-RhoGEF knockdown on vinculin (right), as assessed by immunostaining for each molecule. **c** Effects of PDZ-RhoGEF knockdown on the distribution of F-actin, E-cadherin and MHC-IIA in HT29 cells. Boxed area is enlarged at the right. The ratio of linear to fragmentary junctions was calculated. n = 3272 junctions in 93 colonies for CTRL si; and 1706 junctions in 97 colonies and 1973 junctions in 83 colonies for PDZ-RhoGEF si #1 and #2, respectively, pooled from two independent experiments. **d** Effects of LIMK1 knockdown or LIMKI/cofilin (CFL) double knockdown on the distribution of F-actin, E-cadherin and MHC-IIA in HT29 cells. Boxed areas are enlarged at the right. The ratio of linear to fragmentary junctions was calculated. n = 3474 junctions in 90 colonies for CTRL si; 2964 junctions in 97 colonies and 3058 junctions in 88 colonies for LIMK1 si #1 and #2, respectively; and 2381 junctions in 101 colonies and 3224 junctions in 89 colonies for LIMK1 si #1/cofilin si #1 and LIMK1 si #1/cofilin si #2, respectively, pooled from two independent experiments. **e** Effects of PDZ-RhoGEF/cofilin double knockdown on the distribution of F-actin, E-cadherin and MHC-IIA in HT29 cells. Boxed area is enlarged at the right. The ratio of linear to fragmentary junctions was calculated. n = 1696 junctions in 95 colonies for PDZ-RhoGEF si #1; and 2009 junctions in109 colonies and 2429 junctions in 105 colonies for PDZ-RhoGEF si #1/cofilin si #1 and PDZ-RhoGEF si #1/cofilin si #2, respectively, pooled from two independent experiments. **f** Effects of vinculin/cofilin double knockdown on the distribution of αE-catenin in HT29 cells. Images in **c**, **d**, and **e** were obtained with Airyscan. Scale bars, 10 μm (**a**, **b**, **c** left, **d** left, **e** left, and **f**) and 5 μm (**c** right, **d** right, and **e** right). See also Supplementary Fig. 7

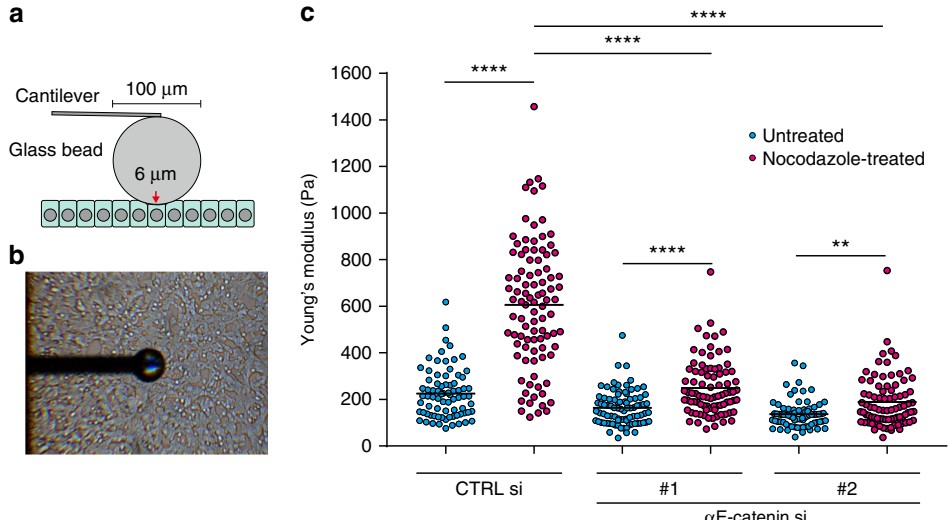

**Fig. 8** Measurement of the apical surface stiffness by atomic force microscopy. **a** Diagrammatic outline of the method. **b** An example of the experiment. **c** The stiffness of cells, treated or not treated with nocodazole, at the apical region. $n = 77$ for CTRL si/untreated, 95 for CTRL si/treated, 86 for αE-catenin si #1/untreated, 86 for αE-catenin si #1/treated, 79 for αE-catenin si #2/untreated, and 84 for αE-catenin si #2/treated. The data were pooled from two independent experiments. **$P = 0.0011$, ****$P < 0.0001$ by two-tailed Mann–Whitney $U$-test. Black lines show mean

0.1% Triton X-100 in PBS for 10 min at room temperature, rinsed in PBS for 5 min at room temperature and blocked in 1.5% bovine serum albumin (BSA) in PBS for 60 min at room temperature. The cells were treated with monoclonal anti-α-tubulin antibody (1:1000) diluted in 1.5% BSA in PBS overnight at 4 °C. After rinsing the cells with PBS, the cells were incubated at room temperature for 60 min in the dark with 1.5% BSA containing goat anti-mouse IgG H&L Alexa Fluor 568 (1:1000). After rinsing the cells in PBS, images of the cells were captured with the Opera imaging system using a ×40 water immersion objective with each channel; Ex1Cam2 = α-tubulin using 561 nm laser excitation and 600/40 nm emission filters and Ex2Cam3 = bright filed using 690 nm TLED and 690/70 nm emission filters. The images were analyzed using Columbus.

**Cell stiffness measurement.** Cell surface stiffness was examined by indentation assay using an AFM cantilever as described previously[55, 56]. Briefly, force probe was prepared by attaching a glass bead (ca. 100 μm diameter) to a tipless silicon cantilever (450 μm long, 50 μm wide, 2 μm thick; nominal spring constant 0.02 N/m; TL-CONT, Nanosensors) using two component Araldite epoxy glue. The cells were set on a plastic culture dish, and pressed from the apical side with a force of 20 nN (the approach and retraction speeds were set to 1.0 μm/s) by the force probe. The measurement using a contact mode was carried out with the Nano-Wizard system with the Cell-Hesion module (JPK), and the data analysis was done with the JPK data processing software (JPK Instruments), where the cell elasticity (Young's modulus) in the apical region was estimated from a force–distance curve using the Hertz model. The cells were cultured in Leibovitz's L-15 medium (Thermo Fisher). Nocodazole treatments were given for 60 min at 37 °C before the measurements. All measurements were performed at room temperature.

**Quantitative analysis of data.** To quantify ZO-1–positive areas, colonies consisting of 30–40 cells were immunofluorescently stained for ZO-1. ZO-1–positive areas were measured using ImageJ (NIH). Fluorescence signals on the images were converted into binary images with "Threshold". The positive areas were measured with "Measure", and divided by total apical areas. The ratio of the values thus obtained from treated cells to those from untreated cells is shown in bar graphs. To quantify the directionality of the retraction of Lifeact-EGFP–labeled protrusions, which is induced by laser ablation, we measured the distances of the retraction from the irradiated point, which generally occurred towards both the cortical and junctional sides. And the ratio of the distances obtained from the opposite sides was calculated for each experiment. For estimating the ratio of linear and fragmentary junctions, we manually evaluated the junctions from morphological criteria.

**Antibodies and reagents.** Rabbit anit-GEF-H1 was kindly provided by Dr. H. Miki[57] (1:500 for immunofluorescence, IF; 1:1000 for western blotting, WB). Rat anti-ROCK1[58] (1:200 for IF), mouse anti-E-cadherin (HECD1, 1:500 for IF, 1:1000 for WB)[59], and rat anti-E-cadherin (ECCD2, 1:500 for IF)[60] were produced previously. The following commercial antibodies were used: Rabbit anti-myosin light chain2 (Cat#3227, 1:100 for WB), rabbit anti-phospho-myosin light chain2 (Cat#3671, 1:50 for IF, 1:1000 for WB), mouse anti-phospho-myosin light chain2 (Cat#3675, 1:50 for IF), rabbit anti-cofilin (Cat#3312, 1:400 for WB and Cat#5175, 1:400 for IF), produced by Cell Signaling; rabbit anti-α-catenin (Cat#C2081, 1:1000

for IF, 1:500 for WB), rabbit anti-β-catenin (Cat#C2206, 1:1000 for IF), mouse anti-α-tubulin (Cat#T9026, 1:1000 for IF, 1:3000 for WB), mouse anti-acetylated tubulin (Cat#T6793, 1:1000 for WB), rabbit anti-myosin IIA (Cat#M8064, 1:1000 for IF, 1:1000 for WB), mouse anti-vinculin (Cat#V4504, 1:200 for IF, 1:500 for WB or Cat#V9131, 1:200 for IF), rabbit anti-l-afadin (Cat#A0349, 1:1000 for IF), produced by Sigma; rabbit anti-ZO-1 (Cat#HPA001636, 1:200 for IF, 1:500 for WB), rabbit anti-ROCK1 (Cat#HPA007567, 1:200 for WB), rabbit anti-PDZ-RhoGEF (Cat#HPA001126, 1:200 for IF, 1:500 for WB), rabbit anti-GRAF2 (Cat#HPA039589, 1:200 for WB), rabbit anti-LIMK1 (Cat#HPA028516, 1:500 for IF), produced by ATLAS; mouse anti-RhoA (Cat#sc-418, 1:100 for IF, 1:100 for WB), mouse anti-GAPDH (Cat#sc-32233, 1:500 for WB), produced by Santa Cruz; rabbit anti-ZO-1 (Cat#HP9043, 1:100 for IF), produced by Hycult Biotech; mouse anti-p190A-RhoGAP (Cat#610149, 1:200 for WB), mouse anti-p190B-RhoGAP (Cat#611612, 1:200 for WB), produced by BD; rabbit anti-Par3 (Cat#07-330, 1:50 for WB) produced by Millipore; mouse anti-RhoA (Cat#ARH03, 1:100 for WB) produced by Cytoskeleton. Uncropped western blots obtained using these antibodies are shown in Supplementary Fig. 8. F-actin was labeled using with Alexa Fluor 488 or 568–labeled phalloidin (Thermo Cat#A12379 or A12380, 1:100).

The following commercial regents were used; nocodazole (Cat#M1404), paclitaxel (Cat#T7402), podophyllotoxin (Cat#P4405), Blebbistatin (Cat#B0560), and mitomycin C (Cat#M0503) produced from Sigma; and Y-27632 (Cat#688000), from Merck. Insoluble reagents were prepared in Dimethyl sulfoxide (DMSO). Upon their administration to cell cultures, DMSO was diluted with culture medium to make its final concentration 0.1%. Nocodazole was used at the concentration of 10 μM, unless otherwise specified.

**DNA constructs.** A DNA expression plasmid of pCA-MLC2-EGFP was described previously[58]. Lifeact-EGFP[61] was generated by inserting its corresponding oligo DNA into a pCAH-EGFP vector. A RhoA-specific RaichuEV, which was designed according to a previous publication[26], was provided by M. Matsuda. pCMV-EGFP-ZO-1[62] was provided by F. Matsuzaki. pCMV-Cofilin S3A-YFP and pCMV-Cofilin S3E-YFP[34] were provided by K. Mizuno.

**siRNA.** Multiple siRNAs were prepared for each molecule. We generally obtained similar results using these siRNAs, and therefore presented data acquired with a single siRNA as a representative in many cases. The following siRNAs were purchased from Sigma-Aldrich: RhoA si #1 (SASI_Hs02_00332284), GRAF2 si #1 (SASI_Hs02_00200016) and si #2 (SASI_Hs02_00356968), 190 A si #1 (SASI_Hs02_00337730), 190B si #1 (SASI_Hs01_00188461), MHC-IIA si #1 (SASI_Hs01_00197338) and si #2 (SASI_Hs01_00197339), PDZ-RhoGEF si #1 (SASI_Hs01_00113112) and si #2 (SASI_Hs02_00346284), E-cadherin si #1 (SASI_Hs01_00086310) and si #2 (SASI_Hs01_00086312), αE-catenin si #1 (SASI_Hs01_000863129) and si #2 (SASI_Hs01_00217323), vinculin si #1 (SASI_Hs02_00335533), LIMK1 si #1 (SASI_Hs01_00154610) and si #2 (SASI_Hs01_00154612), and cofilin1 si #1 (SASI_Hs01_00078353). siRNAs, RhoA si #2 (HSS100655), 190 A si #2 (HSS104485), 190B si #2 (HSS100675), vinculin si #2 (HSS111260), cofilin1 si #2 (HSS173851), and GEF-H1 si #1 (5′-CAGUGAGCUGAUGAGUGACUUUGAG-3′/5′-CUCAAAGUCACUCAU-CAGCUCACUG-3′) and si #2 (5′-GAGACAAACGCUUCCAGCAAUUCAU-3′/

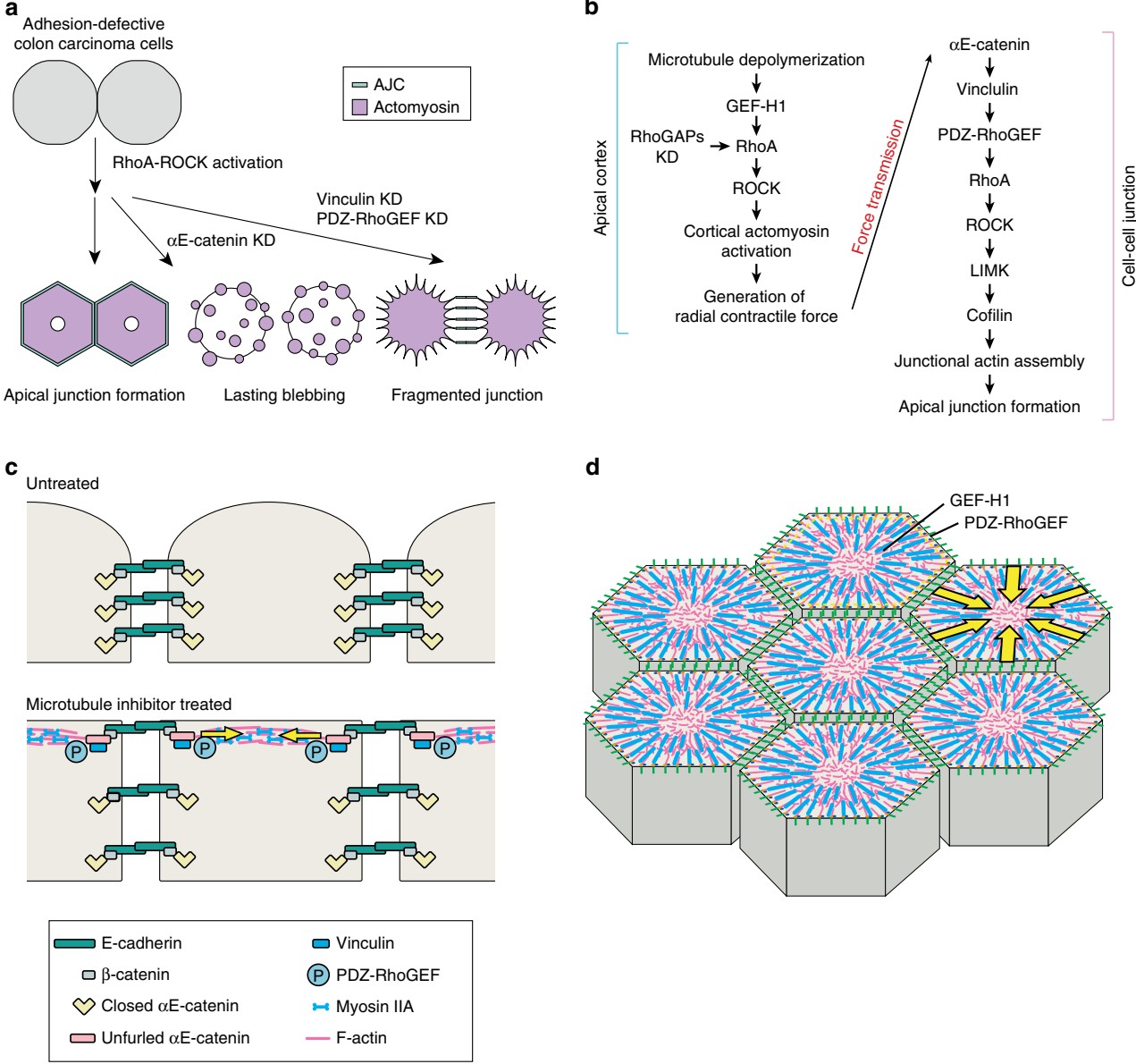

**Fig. 9** Summary and models. **a** Responses of HT29 cells to various treatments. **b** Molecular pathways for AJC induction in HT29 cells. **c**, **d** Molecular events enabling AJCs restoration in adhesion-defective colon carcinoma cells

5′-AUGAAUUGCUGGAAGCGUUUGUCUC-3′), were purchased from Thermo Fisher. We used Stealth RNAi Negative Control Low GC Duplex #2 (Thermo Fisher, Cat#12935110) as a control siRNA (CTRL si).

**Cell culture and transfection**. HT29 (ATCC), Caco2 (ATCC), Colo205[63] (ATCC), LS180 (ATCC), and SW620 (ATCC) cells were cultured in a 1:1 mixture of DMEM and Ham's F12 (Wako) supplemented with 10% FBS. The cells were treated with reagents at 1 day after plating on cover glasses. HT29 cells were transfected with cDNA using Lipofectamine LTX and Plus Reagent (Thermo Fisher), or transfected with siRNA using Neon Transfection System (Thermo Fisher). The cells were used for experiments at 3–5 days after siRNA transfection. Stable transfectants of EGFP-ZO-1, Cofilin S3E-YFP and Cofilin S3A-YFP were selected using with G418. For Matrigel cultures of Colo205 cells, these were plated on cover glasses coated with Matrigel (Corning, Cat#356230), and cultured in medium containing 2% Matrigel for 3 days. For co-culturing HT29 and Caco2 cells, these cells were plated together on a Transwell permeable support (Corning, Cat#3401), and cultured for 2 days. Mycoplasma contamination was checked by DAPI staining.

**Rho GTPase activation assay**. Rho GTPase activation assay was performed as previously described[64], with some modifications. GST-Rhotekin RBD was synthesized in BL21 E. coli, and purified with glutathione magnetic beads (Pierce). Cells were washed with cold TBS, and lysed in a lysis buffer (50 mM Tris-HCl pH

7.2, 1% Triton X-100, 0.1% SDS, 500 mM NaCl, 10 mM MgCl₂ and a protease inhibitor cocktail (Roche)). After centrifugation at 20,000×g for 10 min, the supernatants were incubated with GST-Rhotekin RBD binding beads for 45 min at 4 °C. The beads were washed with a wash buffer (50 mM Tris-HCl pH 7.2, 1% Triton X-100, 150 mM NaCl, 10 mM MgCl₂ and a protease inhibitor cocktail), and boiled in SDS sample buffer.

**Microtubule sedimentation assay**. Microtubule sedimentation assay was performed as previously described[65] with some modifications. Cells were plated on 12-well multi-dishes, and cultured for 1 day. After removing culture medium, cells were lysed with a microtubule stabilizing buffer (100 mM PIPES pH 6.8, 1 mM MgSO₄, 1 mM EDTA, 2 M glycerol, 0.1% (w/v) Triton X-100 and a protease inhibitor cocktail) for 20 min at 37 °C. The lysate was centrifuged at room temperature for 5 min at 16,000×g. After collecting the supernatant as the soluble fraction, the pellet was lysed with a whole cell lysis buffer (10 mM Tris-HCl pH 7.5, 2 mM EDTA, 1% SDS and a protease inhibitor cocktail), and boiled for 15 min. This lysate was further centrifuged at room temperature for 5 min at 20,000×g, and used as the insoluble fraction.

**Immunostaining and microscopy**. For immunostaining proteins, cells were fixed with 1% (w/v) paraformaldehyde in HBSS at 37 °C for 10 min, and permeabilized with 0.25% (w/v) Triton X-100 for 5 min, unless otherwise specified. For

immunostaining of α-tubulin and GEF-H1, cells were fixed, and permeabilized with methanol at −20 °C for 2 min. For immunostaining of RhoA, cells were fixed with 10% TCA at 4 °C for 15 min, and permeabilized with 0.25% (w/v) Triton X-100 for 5 min. The fixed cells were blocked with 3% (w/v) BSA in TBS-T at room temperature for 30 min. The blocked cells were incubated with primary antibodies in Can Get Signal Solution A (TOYOBO) at room temperature for 1 h, and subsequently with secondary antibodies in Can Get Signal solution A (TOYOBO) for 45 min. Coverslips were mounted in FluoroSave (Calbiochem). Images were obtained by using a laser-scanning confocal microscope LSM780 (Carl Zeiss) equipped with PMTs and a GaAsP detector, or LSM880 (Carl Zeiss) equipped with GaAsP detector and Airyscan. Both microscopes were equipped with a Zeiss 49 filter, a Zeiss 38HE-Endow GFP filter, a Zeiss 43HE-Cy3, an alpha-Plan-APOCHROMAT ×100/1.46 oil lens, a Plan-APOCHROMAT ×63/1.4 lens, a Plan-APOCHROMAT ×40/1.3 oil lens or a Plan-APOCHROMAT ×20/0.8 lens. Super-resolution images were obtained using Airyscan of LSM880. Structured illumination microscopy (SIM) images were obtained using ELYRA PS.1 (Carl Zeiss), which was equipped with a EM-CCD camera (iXon885, Andor) and with an alpha-Plan-APOCHROMAT ×100/1.46 oil lens. Images were processed using ZEN (Carl Zeiss). Fluorescence intensity was measured using ImageJ (NIH).

**Laser ablation**. Laser ablation of cells was performed using a FluoView FV1000 laser-scanning confocal microscope (Olympus) equipped with UV-ASU-P2 (Olympus) through a UPLSAPO ×60/1.35 NA oil lens. Images were acquired at every 1 s. 349-nm UV laser was applied to cells at a specific point. The cells were subjected to laser ablation at 1 h after nocodazole administration. Length of retraction was measured at the images taken immediately after laser application, using Image J (NIH). Ratio of retraction toward the cortex side to that toward the junction side was calculated.

**Time-lapse imaging**. Live imaging of cells was performed using a LCV100 (Olympus) equipped with a UAPO ×40/340 objective lens (Olympus), a LED light source, a DP30 camera (Olympus), differential interference contact (DIC) optical components, and interference filters. Live-cell imaging of MLC2-EGFP or Lifeact-EGFP was obtained using the spinning-disc laser confocal microscope IX71 (Olympus) equipped with CSU-X1 (Yokogawa), a Yokogawa YOKO R485/561 filter, a Neo sCMOS camera (ANDOR), and a UPLSAPO ×60/1.35 oil lens. The cells were set on a ChamlideTC $CO_2$ incubator (Live Cell Instrument) at 37 °C. Live-cell imaging of Raichu probes was performed using a spinning-disc laser confocal microscope IX81 (Olympus) equipped with CSU-W1 (Yokogawa), an ImagEM-1K camera (Hamamatsu), and a PLANAPO N ×60/1.42 oil lens. Images were processed using MetaMorph (Molecular Devices) and ImageJ (NIH).

**Statics and reproducibility**. Statistical analysis was performed using Prism7 (GraphPad Software). The sample sizes used for analysis were similar to those generally employed in the field. Two-tailed Mann–Whitney $U$-test was used for statistical analysis of ZO-1 area and AFM data. All graphs except for ratio data were showed with ±SEM. All experiments were repeated at least two times, except for those to confirm the knockdown efficiency of siRNAs, which was performed only once if the results were consistent with the manufacturers' prediction.

**Data availability**. All data that support the conclusions are available within the Article and Supplementary Files, or available from the corresponding author on request.

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

## Acknowledgments

We thank T. Fukami, S. Fujita, T. Goto, M. Yoshida, A. Tanaka and M. Shirouzu for supports of drug screening; and Chemical Bank Unit for Drug Discovery Platform, RIKEN Center for Sustainable Resource Science, and Drug Discovery Initiative University of Tokyo for chemical libraries. We also thank H. Miki for anti-GEF-H1 antibody; M. Matsuda for FRET probes; F. Matsuzaki for EGFP-ZO-1 plasmid; K. Mizuno for cofilin mutant plasmids; M. Ohgushi and M. Eiraku for AFM and FRET experiments; S. Hayashi and H. Wada for laser ablation experiments; S. Yonemura, T. Otani, M. Furuse and W. Meng for discussion; M. Kawasaki, Y. Inoue, H. Saito, and S. Hiver, for technical supports. The imaging experiments were performed at the Riken Kobe light microscopy facility. This work was supported by a Grant-in-Aid for Specially Promoted Research (grant number 20002009) from the Japan Society for Promotion of Science to M.T., and also by Platform for Drug Discovery, Informatics, and Structural Life Science from the Ministry of Education, Culture, Sports, Science and Technology, Japan.

## Author contributions

S.I. and M.T. designed the study; S.I. performed the majority of experiments; S.O. performed AFM experiments; M.A., M.F., T.O., carried out drug screening; T.N. constructed reagents and performed some experiments; S.I. and M.T. wrote the manuscript.

## Additional information

**Competing interests:** The authors declare no competing financial interests.

