## [Peer Review File · Nature Communications]

Reviewers' comments:

Reviewer #1 (Remarks to the Author):

In this work, Shoko Ito et al., report that cortical tension induced by microtubule depolymerization, restores functional junctions in epithelial cells that express all components of the adherens junctions but are adhesion defective. The work follows several reports from the Takeichi lab and more recently from others, indicating that serine/threonine phosphorylation events and p120 catenin binding negatively regulate adhesion in Colo 205 cells, results that gave rise to the notion of inside-out regulation of cadherin adhesive function. The current work uncovers an interesting relationship between the microtubules and RhoA signaling in these events. First, the authors show that microtubule depolymerizing drugs promote the compaction and adhesion of HT29, Colo 205 and other cell lines. While interesting, this effect of microtubule depolymerizers has already been reported recently in PLoS One. However, the authors of this study go on to show that GEF-H1, which is inhibited by binding to microtubules, promotes RhoA activation in these cells upon treatment with nocodazole, resulting in the cortical recruitment of myosin II and the contraction of the apical cortex. In these cells, MHC IIA does not localize to the junctions, but to an apical sarcomeric structure, that links radially to the apical adherens complexes and exerts force. Force results in recruitment of vinculin, as reported by others, and the concomitant recruitment of PDZ-RhoGEF to the junctions. This latter event is linked to junction stabilization and restoration of adhesion in these carcinoma cell lines.

The novelty of this work is not so much on the microtubule depolymerization effects, but on the RhoA/ROCK mediated force-induced stabilization of adherens junctions. To what extent this signaling links to p120 phosphorylation and cadherin inside-out signaling is not interrogated in this work, although others have shown that nocodazole promotes p120 dephosphorylation. A link of this work to prior reports on p120 catenin would strengthen overall impact. For example, the PI has shown before that p120-uncoupled E-cadherin can rescue adhesion in Colo 205 cells. Is this effect associated with increased cortical tension and recruitment of vinculin and PDZ-RhoGEF to the junctions?

There is also an attempt by the authors to link this pathway to cancer cell migration and metastasis. Frankly, this is a very tenuous relationship that will have to be tested in depth before one can even speculate. For example, while the authors argue that restoring functional junctions may limit dissemination, several studies have shown that collective cell invasion depends on cell-cell junctions, while cells at the leading edges of collectively migrating cells exhibit high RhoA activity. Furthermore, single cells disseminating in an amoeboid fashion have increased RhoA activation.

An important issue to be resolved here is whether the cortical structures induced by microtubule depolymerization are also seen in normal or non-transformed epithelial monolayers. Is radial apical tension a key part of epithelial organization lost in carcinoma cells, or is it a side effect of the artificial depletion of microtubules? The data showing that depleting RhoGAPs phenocopies these effects probably argues that the overall activity level of RhoA is important rather than GEF-H1 and the organization of the microtubules per se. In RhoGAP depleted cells, what happens to GEF-H1 localization, and do RhoA/ROCK relocalize to the junctions?

Finally experiments with cofilin mutants and LIMK are interesting, however, the authors should also describe how these mutants affect cortical tension and actomyosin organization, expected maybe to be affected, and not only effects on the junctions.

In short, this is an interesting report that builds on prior work by this and other groups. To be impactful, the authors need to clarify whether the cortical sarcomeric structure and resulting tension is essential for normal epithelial function, or is a specific trait of carcinoma cells, and relate this work to p120 catenin.

Reviewer #2 (Remarks to the Author):

The authors address a key question at the boundary of cell and cancer biology—how do cell-cell junctions and the cytoskeleton interact and how is this disrupted in cancer. We have known for more than 20 years that most metastatic epithelial tumors have major alterations in cell adhesion, but that only a subset of these have mutations in or downregulation of core adherens junction components. Further, over that same period, a number of labs have published examples of how certain broad spectrum inhibitory compounds like the PP2 inhibitor could alter junctional architecture in certain cancer cells, but the underlying mechanism have remained illusive. Here the Takeichi lab presents a masterful and comprehensive set of experiments that alter that picture. Starting with an impressive chemical genetic screen, they find that microtubule depolymerization can dramatically restore epithelial cell junction and cytoskeletal architecture in a number of different colorectal cancer cell lines. They then follow this lead in an extremely impressive way, defining a molecular pathway linking a Rho-GEF step by step through an elaborate pathway to junctional reassembly. The data are gorgeous, the conclusions well-supported, and the implications very important. I think this work will be of broad interest to cell, developmental and cancer biologists. I had some comments and suggestions that I think will strengthen the manuscript and would support publication of a revised manuscript.

p. 4. Could the authors clarify what they meant by stating that they found “55 compounds (that) exhibited structures that could potentially inhibit microtubule polymerization”. Also, in Suppl. Table 1, when showing an “analog”, it would be helpful to also show the structure of the compound itself.

Fig 1h and 2i. The differences between control and experimental values are clearly different, but I was surprised at the number of examples in both in which there was “no lumen” – how do the authors interpret this? I’d also suggest altering the wording from “they organized fully polarized cysts when treated with nocodazole, otherwise they formed irregular spheres with multiple lumens” to “nocodazole treatment promoted formation of fully polarized cysts rather than irregular spheres with multiple lumens”—this is how they describe the results in Fig. 2i.

Fig 2A. While the quantitation was clear, a more representative blot would be helpful. I also think the authors should more clearly make the point that the major difference is not a global increase in Rho activity but rather a local activation of Rho at nascent cell junctions.

Fig 3d. The authors present awesome super-resolution images of actin and myosin organization after nocodazole treatment. However, I was puzzled by the description as “sarcomeric”—it seemed to me that the orientation of the myosin and actin filaments was not right for that, in contrast, for example, to what the Kachar and Peifer labs observed recently at cell-cell junctions in some cell types.

Fig 5A. These actin protrusive structures were exceptionally interesting—I’d suggest adding inset closeups to more clearly show them.

Fig 5b Could the authors more clearly define the units on the Y axis?

Fig 5c—I thought the junctional re-arrangements after vinculin knockdown were very cool—I would like the authors to describe this effect in more detail.

Fig 6d—These experiments were the only ones in the paper I thought were relatively weak—the effects were somewhat modest. Given the breath and depth of the whole analysis, I wonder if these details are essential, or if, perhaps, these particular conclusions should be softened.

For me among of the most exciting aspects of this work were the implications for the cross-talk between cell-junctions and the cytoskeleton. I would suggest adding to the discussion a more

comprehensive description of this interplay. They could also contrast it to things observed in other cell types, where similar cross talk is seen but the consequences for junctional and cytoskeletal architecture are distinct—e.g., Ebrahim et al., 2013, Leerberg et al., 2014, and Choi et al., 2016. The editors also might look into the status of a paper in revision from the Matter lab at Nature Cell Biology that takes a completely different approach to exploring the regulation of the apical myosin cytoskeleton in the apical domain—if it is in press discussing its results would also add interest.

Reviewer #3 (Remarks to the Author):

The authors describe how epithelial cells from cancer lineage can control their adherens junctions specifically by regulating the microtubule machinery. They have used a high content screening approach to identify the microtubule network to be important. This surprising result is thoroughly investigated by a variety of in vitro experiments. And although the authors have the specific chemical compounds, they did not test their hypothesis in an animal model. Since the finding is significant and potentially very interesting and useful, it feels like a missed opportunity not to include such an in vivo experiment. In addition to that, I miss in vitro functionality assays that can tell that the junctions are indeed improved in a functional manner using a resistance or permeability assay. Nevertheless, the work in general is elegantly done and would add to the current knowledge of how tension works on junctions. I have some additional comments that I have listed below.

1. To show that nocodazole improves functional junctions, the authors should include resistance/permeability measurements. This should be done throughout the paper for several proteins that they include in their signaling pathway, eg. GEF-H1, ROCK. Only showing IF is not enough to convincingly show the junctional effects of nocodazole.
2. Figure 2: the Rho staining, based on the antibody staining misses controls. If the authors wish to show the distribution of Rho, please control the Ab specificity with a knock down approach. And check if the localization is similar to the GFP-tagged wt Rho constructs.
3. I am not impressed by the FRET data. Please improve those or repeat with a sensor that has a higher dynamic range. The data presented in SFig2a are of poor quality.
4. Is ROCK protein expression increased upon nocodazole? SFig2B suggests so, an increase in cytoplasm as well as junctional. Please confirm with Western blotting.
5. Second paragraph page 6: "...upregulated Rho..." . I guess the author mean activation rather than protein expression? Fig2A shows equal RhoA protein levels. Next paragraph, same thing.
6. Fig 4f: "...became condensed but in an irregular fashion..." ; this is difficult to judge. The authors should somehow quantify this.
7. For some pics, it might be appropriate to include zooms to improve clarification, like for several merges in fig 5c-d and 6b-c.
8. For the laser ablation, only checking alpha-ctn knock down is a bit minor. I would prefer to see data when vinculin is silenced. The authors have the tools to perform the experiment.
9. I appreciate figure 8. In scheme b, the authors propose a dual activation of RhoA. Do they suggest that there are two different pools of RhoA being involved here, or can the same molecule be activated twice in their pathway? I would like to hear the authors' opinion on that. I would suggest to add a small piece on local and specific signaling to the discussion.

Minor:

- Throughout the Ms, control is spelled as CNTRL whereas CTRL is more common.
- Fig4d: please choose more distinct colors

Response to the Reviewers

We would like to thank the reviewers for their careful reading of our manuscript and thoughtful comments. Our point-by-point responses are below. The original reviews are in italic. Since the original Figure 2 was split in two, the figure numbers were changed accordingly.

Reviewer #1

The novelty of this work is not so much on the microtubule depolymerization effects, but on the RhoA/ROCK mediated force-induced stabilization of adherens junctions. To what extent this signaling links to p120 phosphorylation and cadherin inside-out signaling is not interrogated in this work, although others have shown that nocodazole promotes p120 dephosphorylation. A link of this work to prior reports on p120 catenin would strengthen overall impact. For example, the PI has shown before that p120-uncoupled E-cadherin can rescue adhesion in Colo 205 cells. Is this effect associated with increased cortical tension and recruitment of vinculin and PDZ-RhoGEF to the junctions?

Response: We appreciate this comment. Actually, we have been interested in the problem of how the present findings are related to the previous observation that p120-catenin (p120) uncoupled cadherins can also restore the adhesions between Colo205 cells. Curiously, exogenous expression of such mutant cadherins did not show adhesion-promoting effects when HT29 cells were used. Therefore, we could not compare the roles of p120-uncoupled cadherin expression with those of RhoA activation on cell adhesion using HT29 cells. We suspect that physiological conditions required for responding to p120-uncoupled cadherins may differ between cell lines. Nevertheless, the role of p120 is an interesting issue, and therefore we have added discussion of how p120 function could be related to RhoA activation in promotion of cell-cell adhesion to the second paragraph of Discussion (pages 14-15).

There is also an attempt by the authors to link this pathway to cancer cell migration and metastasis. Frankly, this is a very tenuous relationship that will have to be tested in depth before one can even speculate. For example, while the authors argue that restoring functional junctions may limit dissemination, several studies have shown that collective cell invasion depends on cell-cell junctions, while cells at the leading edges of collectively migrating cells exhibit high RhoA activity. Furthermore, single cells disseminating in an amoeboid fashion have increased RhoA activation.

Response: We agree that the potential linkage between the ability of cells to disperse and cancer metastasis needs more vigorous tests *in vivo*. Therefore, we discussed this problem more carefully, choosing more appropriate wording (e.g., cancer dissemination instead of cancer metastasis).

An important issue to be resolved here is whether the cortical structures induced by microtubule depolymerization are also seen in normal or non-transformed epithelial monolayers.

Response: We have never observed that microtubule depolymerization induces changes in cortical actomyosin in the epithelial cells with normal architecture, such as Caco2 and MDCK, whether they are transformed or not. Therefore, we believe that the microtubule depolymerization-dependent reorganization of actomyosin, reported in this manuscript, occur specifically in a group of advanced carcinoma cells that have lost the original epithelial architecture. We re-emphasized this point throughout the manuscript comparing the responses of Caco2 cells to nocodazole with that of HT29.

Concerning the data on MDCK, which is considered a non-transformed line, we did not include them in the manuscript, as this cell line was derived from kidney, whereas all other cell lines were from colon. We are afraid that tumor cells of different origins might behave differently, and therefore we restricted our analysis to colon-derived cells. Since there are no 'non-transformed cell lines', which are derived from the colon, we used Caco2 cells as an example of epithelial cells with normal epithelial architecture and junctions. If readers are interested in whether our model is applicable to non-transformed epithelial cells, they can easily test the effects of nocodazole on various epithelial cells such as MDCK in their own laboratory.

Is radial apical tension a key part of epithelial organization lost in carcinoma cells, or is it a side effect of the artificial depletion of microtubules? The data showing that depleting RhoGAPs phenocopies these effects probably argues that the overall activity level of RhoA is important rather than GEF-H1 and the organization of the microtubules per se.

Response: We think that the overall activity of RhoA is most crucial, and the microtubule depolymerization-dependent activation of GEF-H1 was merely a way to activate RhoA. To support this idea, we added a novel result showing that GEF-H1 knockdown did not significantly interfere with the adhesion-inducing effect of RhoGAPs depletion (new Fig. 3b). We also discuss this problem more thoroughly in the first paragraph of the revised Discussion (page 14).

In RhoGAP depleted cells, what happens to GEF-H1 localization, and do RhoA/ROCK relocate to the junctions?

Response: GEF-H1 tended to increase at apical regions even in RhoGAPs-depleted cells, where microtubules remained polymerized, suggesting that its apical increase may not be relevant to its functional activation, which probably occurred according to overall changes in cell polarity after AJC formation. Furthermore, as commented above, GEF-H1 was not necessary for AJC recovery in RhoGAPs-depleted cells. Thus, to avoid non-essential discussion, we decided to remove the result of the GEF-H1 distribution along the vertical axis. Concerning RhoA/ROCK1 distribution, both increased in the apical cortex and junctions in RhoGAP-depleted cells. We added this result to Supplementary Fig. 3b.

Finally experiments with cofilin mutants and LIMK are interesting, however, the authors

should also describe how these mutants affect cortical tension and actomyosin organization, expected maybe to be affected, and not only effects on the junctions.

Response: As seen in new Fig. 7d, myosin IIA remains condensed in LIMK1-depleted cells. This is also the case in S3A-cofilin expressing cells (data not shown). These suggest that LIMK1-cofilin signaling is not involved in cortical actomyosin organization. We have clearly described these observations on pages 12-13.

In short, this is an interesting report that builds on prior work by this and other groups. To be impactful, the authors need to clarify whether the cortical sarcomeric structure and resulting tension is essential for normal epithelial function, or is a specific trait of carcinoma cells, and relate this work to p120 catenin.

Response: Thank you for the constructive comments. All these points have been addressed above. Especially, we have added our view of how previous observations about p120-catenin are related to the present findings, to the second paragraph of Discussion, as mentioned above.

Reviewer #2

p. 4. Could the authors clarify what they meant by stating that they found “55 compounds (that) exhibited structures that could potentially inhibit microtubule polymerization”. Also, in Suppl. Table 1, when showing an “analog”, it would be helpful to also show the structure of the compound itself.

Response: Thanks for this comment. We realized that this point was mis-described in the original text. In fact, these 55 compounds inhibited microtubule polymerization in our screening, although they were not compounds registered as microtubule inhibitors. Therefore, we re-wrote this part. Concerning ‘analog’, we have now shown the structure of the compound itself other than its analog in Supplementary Table 1, except for the compounds for which only analogs were hit in our screening.

Fig 1h and 2i. The differences between control and experimental values are clearly different, but I was surprised at the number of examples in both in which there was “no lumen” – how do the authors interpret this? I’d also suggest altering the wording from “they organized fully polarized cysts when treated with nocodazole, otherwise they formed irregular spheres with multiple lumens” to “nocodazole treatment promoted formation of fully polarized cysts rather than irregular spheres with multiple lumens”—this is how they describe the results in Fig. 2i.

Response: Many cells aggregated without forming any lumen. We classified these cell aggregates as ‘no lumen’. Referring to the recommendation by the reviewer, we have rewritten this part, hoping that it is now improved.

Fig 2A. While the quantitation was clear, a more representative blot would be helpful. I also think the authors should more clearly make the point that the major difference is not a global increase in Rho activity but rather a local activation of Rho at nascent cell junctions.

Response: We believe that the blot shown in Fig. 2a is a representative one. Concerning the problem of Rho activation sites, we have more carefully discussed this point within the first paragraph of Discussion (page 14), also responding to a comment by Reviewer #3. Our results rather suggest that a global increase in RhoA activity is sufficient to promote adhesion.

Fig 3d. The authors present awesome super-resolution images of actin and myosin organization after nocodazole treatment. However, I was puzzled by the description as “sarcomeric”—it seemed to me that the orientation of the myosin and actin filaments was not right for that, in contrast, for example, to what the Kachar and Peifer labs observed recently at cell-cell junctions in some cell types.

Response: Indeed, the orientation of myosin IIA and actin filaments, which are reorganized in nocodazole-treated cells, is not as simple as that observed along cell junctions by these authors, particularly because the actin filament orientation is complex. We have discussed this point more carefully on page 15, and also changed the cartoon drawing myosin IIA and actin filaments in new Fig. 9d, which now summarizes our observations with greater precision than in the previous version.

Fig 5A. These actin protrusive structures were exceptionally interesting—I’d suggest adding inset closeups to more clearly show them.

Response: We have added a set of close-ups to the new Fig. 6a.

Fig 5b Could the authors more clearly define the units on the Y axis?

Response: We have changed the figure and legend for clearer presentation of the data (new Fig. 6b).

Fig 5c—I thought the junctional re-arrangements after vinculin knockdown were very cool—I would like the authors to describe this effect in more detail.

Response: To help readers understand how junctional proteins are rearranged after vinculin depletion, we have added enlarged images to the new Fig. 6d.

Fig 6d—These experiments were the only ones in the paper I thought were relatively weak—the effects were somewhat modest. Given the breath and depth of the whole analysis, I wonder if these details are essential, or if, perhaps, these particular conclusions should be softened.

Response: We would like leave Fig. 7d (originally 6d) to logically discuss the potential involvement of LIMK and cofilin. On the other hand, factors other than LIMK/cofilin may also be involved in vinculin-dependent re-enforcement of junctions. Therefore, we discussed this possibility more thoroughly than before, on pages 15 to 16.

For me among of the most exciting aspects of this work were the implications for the cross-talk between cell-junctions and the cytoskeleton. I would suggest adding to the discussion a more comprehensive description of this interplay. They could also contrast it to things observed in other cell types, where similar cross talk is seen but the consequences for junctional and cytoskeletal architecture are distinct—e.g., Ebrahim et al., 2013, Leerberg et al., 2014, and Choi et al., 2016.

Response: We thank the reviewer for this comment. We have extended the discussion section for discussing the interactions between cell junctions and actomyosin in more detail, referring to earlier works that were listed by the reviewer, particularly on pages 15 to 16.

The editors also might look into the status of a paper in revision from the Matter lab at Nature Cell Biology that takes a completely different approach to exploring the regulation of the apical myosin cytoskeleton in the apical domain—if it is in press discussing its results would also add interest.

Response: We could not identify the work suggested here.

Reviewer #3 (Remarks to the Author):

And although the authors have the specific chemical compounds, they did not test their hypothesis in an animal model. Since the finding is significant and potentially very interesting and useful, it feels like a missed opportunity not to include such an in vivo experiment.

Response: Thanks for this comment. We agree that the most exciting step of this study is to test whether altered cell junctions affect the *in vivo* behavior of cancer cells. To do this, we need to prepare tumor cells whose junctions are stably rescued without using microtubule inhibitors, as these inhibitors are poisonous for animals in effective concentrations. We are actually designing such cells, but it will take considerable time to obtain them. We believe that the current cell biological data are sufficiently novel to publish, and we would like to leave *in vivo* analysis for the next cycle of this study.

1. To show that nocodazole improves functional junctions, the authors should include resistance/permeability measurements. This should be done throughout the paper for several proteins that they include in their signaling pathway, eg. GEF-H1, ROCK. Only showing IF is not enough to convincingly show the junctional effects of nocodazole.

Response: As recommended by the reviewer, we have done the resistance/permeability measurements. However, we did not detect any significant increases of these tight junction-dependent parameters after adhesion-inducing treatments. Therefore, we carefully re-analyzed the structures of the reorganized junctions, and noticed that these junctions did not completely cover the entire cell-cell contacts particularly at tri-cellular or multi-cellular junctions, probably causing leaky junctions. For example, please see discontinuity of ZO-1 or cadherin/catenin distribution, which is indicated with an enlargement and arrows in Fig. 1b

and 1f, respectively. Therefore, we now state this incompleteness of junction recovery in the Results (page 5) and Discussion (pages 16 to 17) sections.

2. *Figure 2: the Rho staining, based on the antibody staining misses controls. If the authors wish to show the distribution of Rho, please control the Ab specificity with a knock down approach. And check if the localization is similar to the GFP-tagged wt Rho constructs.*

Response: We show the immunostaining for RhoA in control and RhoA-depleted cells in the new Fig. 2b, confirming that the antibody staining is specific. We also prepared cells in which GFP-RhoA is expressed, finding that it was diffusely distributed in the cytoplasm of control cells, whereas it accumulated at cell peripheries after nocodazole treatment, in addition to the cytoplasm (see below). We believe that the immunostaining data with control and RhoA-depleted cells are sufficient to confirm the specificity of the antibody used, therefore we did not add the GFP-RhoA data to the manuscript.

3. *I am not impressed by the FRET data. Please improve those or repeat with a sensor that has a higher dynamic range. The data presented in SFig2a are of poor quality.*

Response: We replaced the FRET data with a better image.

4. *Is ROCK protein expression increased upon nocodazole? SFig2B suggests so, an increase in cytoplasm as well as junctional. Please confirm with Western blotting.*

Response: We have shown a Western blot for ROCK in Supplementary Fig. 2e. Its protein expression level did not change upon nocodazole treatment.

5. *Second paragraph page 6: "...upregulated Rho..." . I guess the author mean activation rather than protein expression? Fig2A shows equal RhoA protein levels. Next paragraph, same thing.*

Response: The wording "upregulate or upregulation" is indeed confusing. We actually did not find any increase of RhoA protein, as shown in Supplementary Fig. 2b. We have used these words more carefully now.

6. Fig 4f: “...became condensed but in an irregular fashion...” ; this is difficult to judge. The authors should somehow quantify this.

Response: We agree that it was difficult to distinguish between p-MLC2 patterns in control and α E-catenin-depleted cells. Since the most dramatic difference between these cells was observed in myosin IIA distribution rather than p-MLC2, we now show only myosin IIA images in this figure (new Fig. 5f). Note that myosin IIA forms stripes only in control cells.

7. For some pics, it might be appropriate to include zooms to improve clarification, like for several merges in fig 5c-d and 6b-c.

Response: We now show zooms in new Figs. 6d, 7c, 7d, and 7e, as these are particularly important for readers to follow our description.

8. For the laser ablation, only checking alpha-ctn knock down is a bit minor. I would prefer to see data when vinculin is silenced. The authors have the tools to perform the experiment.

Response: There seems to be some confusion here, as we used only control cells, not α E-catenin-depleted cells, in laser ablation experiments. Whatever the case, we followed the reviewer’s suggestion to do laser ablation in vinculin-depleted cells. The results are shown in Figure 6d. Thanks to the reviewer’s recommendation, we have been able to add novel data that strengthen the paper.

9. I appreciate figure 8. In scheme b, the authors propose a dual activation of RhoA. Do they suggest that there are two different pools of RhoA being involved here, or can the same molecule be activated twice in their pathway? I would like to hear the authors’ opinion on that. I would suggest to add a small piece on local and specific signaling to the discussion.

Response: We have added a section to discuss how RhoA works at multiple sites to the first paragraph of the Discussion (page 14).

Minor:

- Throughout the Ms, control is spelled as CNTRL whereas CTRL is more common.

Response: We have changed this point.

- Fig4d: please choose more distinct colors

Response: We have changed the colors.

REVIEWERS' COMMENTS:

Reviewer #1 (Remarks to the Author):

The authors have addressed all my previous concerns, either with new data or with discussion in the revised manuscript. I have no further concerns.

Reviewer #2 (Remarks to the Author):

As I noted in my original review, the authors address a key question at the boundary of cell and cancer biology—how do cell-cell junctions and the cytoskeleton interact and how is this disrupted in cancer. Here the Takeichi lab presents a masterful and comprehensive set of experiments that alter that picture. Starting with an impressive chemical genetic screen, they find that microtubule depolymerization can dramatically restore epithelial cell junction and cytoskeletal architecture in a number of different colorectal cancer cell lines. They then follow this lead in an extremely impressive way, defining a molecular pathway linking a Rho-GEF step by step through an elaborate pathway to junctional reassembly. The data are gorgeous, the conclusions well-supported, and the implications very important. I had some relatively minor issues with the original manuscript, which the authors have fully addressed. I think this work will be of broad interest to cell, developmental and cancer biologists.

Reviewer #3 (Remarks to the Author):

I have read the revised manuscript carefully including the point-by-point reply of the authors. I can say that I am more than satisfied with the answers the authors gave me on my concerns.

Response to the Reviewers

As you see below, the reviewers have not provided any additional comments to further revise the manuscript.

Reviewer #1 (Remarks to the Author):

The authors have addressed all my previous concerns, either with new data or with discussion in the revised manuscript. i have no further concerns

Reviewer #2 (Remarks to the Author):

As I noted in my original review, the authors address a key question at the boundary of cell and cancer biology—how do cell-cell junctions and the cytoskeleton interact and how is this disrupted in cancer. Here the Takeichi lab presents a masterful and comprehensive set of experiments that alter that picture. Starting with an impressive chemical genetic screen, they find that microtubule depolymerization can dramatically restore epithelial cell junction and cytoskeletal architecture in a number of different colorectal cancer cell lines. They then follow this lead in an extremely impressive way, defining a molecular pathway linking a Rho-GEF step by step through an elaborate pathway to junctional reassembly. The data are gorgeous, the conclusions well-supported, and the implications very important. I had some relatively minor issues with the original manuscript, which the authors have fully addressed. I think this work will be of broad interest to cell, developmental and cancer biologists.

Reviewer #3 (Remarks to the Author):

I have read the revised manuscript carefully including the point-by-point reply of the authors. I can say that I am more than satisfied with the answers the authors gave my on my concerns.